# Tart Cherry (Fruit of *Prunus cerasus*) Concentrated Powder (TCcp) Ameliorates Glucocorticoid-Induced Muscular Atrophy in Mice

**DOI:** 10.3390/medicina57050485

**Published:** 2021-05-12

**Authors:** Sae-Kwang Ku, Jong-Min Lim, Hyung-Rae Cho, Khawaja Muhammad Imran Bashir, Young Suk Kim, Jae-Suk Choi

**Affiliations:** 1Department of Anatomy and Histology, College of Korean Medicine, Daegu Haany University, 1, Hanuidae-ro, Gyeongsan-si, Gyeongsangbuk-do 38610, Korea; gucci200@hanmail.net; 2Glucan Corporation, 25-15, Worasan-ro 950beon-gil, Munsan-eup, Jinju-si, Gyeongsangnam-do 52840, Korea; kobuks@glucan.co.kr (J.-M.L.); sr0701@glucan.co.kr (H.-R.C.); 3German Engineering Research and Development Center for Life Science Technologies in Medicine and Environment, 31, Gwahaksandan 1-ro, 60 bean-gil, Gangseo-gu, Busan 46742, Korea; imranagrarian3@gmail.com; 4Department of Food Biotechnology, College of Medical and Life Sciences, Silla University, 140, Baegyang-daero 700beon-gil, Sasang-gu, Busan 46958, Korea

**Keywords:** glucocorticoid, muscular atrophy, tart cherry, *Prunus cerasus*, mice

## Abstract

*Background and Objectives*: The present study investigated the beneficial effects of tart cherry (fruit of *Prunus cerasus*) concentrated powder (TCcp) on glucocorticoid (GLU)-induced catabolic muscular atrophy in the skeletal muscle of mice. Furthermore, its potential mechanism was also studied. *Materials and Methods*: Changes in calf thickness, calf muscle weight, calf muscle strength, body weight, gastrocnemius muscle histology, immunohistochemistry, serum creatinine, creatine kinase, lactate dehydrogenase, and antioxidant defense systems were measured. Malondialdehyde, reactive oxygen species, glutathione content, catalase, and superoxide dismutase activities in the gastrocnemius muscle, and muscle-specific mRNA expressions were evaluated. *Results*: After 24 days, GLU control mice showed muscular atrophy at all criteria of indexes. The muscular atrophy symptoms were significantly inhibited by oral treatment with 250 mg/kg and 500 mg/kg of TCcp through antioxidative and anti-inflammatory modulated expression of genes involved in muscle protein degradation (myostatin, atrogin-1, SIRT1, and MuRF1) and synthesis (A1R, Akt1, TRPV4, and PI3K). *Conclusions*: This study shows that the TCcp (500 mg/kg and 250 mg/kg) could improve muscular atrophies caused by various etiologies.

## 1. Introduction

Sarcopenia (gradual loss of muscle strength and mass) represents an important risk factor for aging [1,2,3]. Disturbances in skeletal muscle mass profoundly affect a patient’s daily life. Reduction in physical activity causes further skeletal muscle atrophy, resulting in a brutal circle of atrophic processes [4]. The factors include denervation, joint immobilization, musculoskeletal injury, joint and ligament injuries, joint inflammation, glucocorticoid treatment, prolonged bed rest, sepsis, aging, and cancer cause muscle atrophy [5,6,7].

Various animal models of skeletal muscle atrophy including, immobilization [8], unloading [9,10], denervation [11], starvation [12], and glucocorticoid administration [13] have been used in previous studies. Among those, high doses of dexamethasone (DEXA, a representative glucocorticoid; GLU) induce catabolic variations in skeletal muscle, mainly due to muscle proteolysis [14,15]. The activation of lysosome and ubiquitin-proteasome mediates the GLU-induced protein degradation [16]. Particularly, the muscle-specific atrogin-1, E3-ligases, and muscle ring finger 1 are highly activated by GLUs [17,18,19]. Furthermore, up-regulation of myostatin (a crucial negative regulator of skeletal muscle mass) is also involved in GLU-induced catabolic muscular atrophy [20,21]. Muscle mass and structure are defined by a balance between protein degradation and synthesis [4,10]. The efficacy of the disused muscular atrophic animal models is estimated by measuring the mRNA expression patterns of these proteins using qRT-PCR (quantitative real-time reverse transcription polymerase chain reaction) [10,22]. Damage in muscular antioxidant defense system and muscle fiber apoptosis also contribute in GLU-induced catabolic striate muscle atrophy [23]. Additionally, GLU-induced skeletal muscle atrophy is widely used as a valuable and fast animal model for testing agents linked with abnormal catabolic muscular atrophy [24].

An orally active steroid, oxymetholone [17-hydroxy-2-hydroxymethylenel-17-methyl-5-androstim-3-one], having a fully saturated A-ring structure, may reduce the risk of osteoporosis [25]. It has shown lower androgenic and higher anabolic activity compared to the testosterone propionate, testosterone, and methyl testosterone [26]. Oxymetholone is approved by the United States Food and Drug Administration (US-FDA) to treat anemia and as a reference drug to develop muscle enhancers [24,27]. However, it has also been linked with severe hepatotoxicity [28,29] and decreased anticoagulant tolerance [30].

Tart cherries (Fruit of *Prunus cerasus* L., Rosaceae) and their byproducts contain numerous phytochemicals including the flavonoids kaempferol, isorhamnetin, quercetin, epicatechin, catechin, anthocyanin, and procyanidins [31,32,33,34,35,36,37]. Tart cherries have shown antioxidative [38,39], anti-inflammatory [30], anti-hypertensive [40], memory impairment lowering [41], and cardio-protection improving properties [30,31]. Furthermore, tart cherry extracts have shown accelerated exercise recovery in animal and human trials [38,39,42]. It is therefore hypothesized that tart cherries could show favorable effects on muscular atrophies from various etiologies. The present study investigated the possible favorable effects of tart cherry concentrated powder (TCcp) on skeletal muscular preservation in the GLU-induced muscular atrophic mouse model.

In the present study, catabolic GLU-induced muscular atrophy was evoked by subcutaneous treatment with DEXA (1 mg/kg; once daily for 10 days). Furthermore, the effects of TCcp on catabolic GLU-induced muscular atrophy and the possible mechanisms were studied for 24 days.

## 2. Materials and Methods

### 2.1. Sample

Pink TCcp powder (Anderson Global Group, Irvine, CA, USA) was dissolved at a concentration of 50 mg/mL in distilled water (as vehicle). In addition, oxymetholone (50 mg tablet; Celltrion Pharm, Incheon, Korea) dissolved at a concentration of 15 mg/mL in distilled water was used as a reference drug. All samples were stored at 4 °C to protect from light and degeneration.

### 2.2. Animal Husbandry and Experimental Design

Sixty adult male Specific Pathogen Free (SPF) Institute of Cancer Research (ICR) mice (Orient Bio, Seungnam, Korea) weighing 27–30 g were allocated to polycarbonate cages (4–5 animals per cage) in humidity (40–45%) and temperature (20–25 °C) controlled conditions. During acclimatization, the light:dark cycle was maintained for 12:12 h; water and normal rodent diet (Purina feed, Seungnam-si, Korea) were supplied ad libitum. After 10 days of acclimatization, excluding overweight and underweight mice, six groups containing eight mice each were assigned as follows: intact control, GLU control (1 mg/kg DEXA), oxymetholone (50 mg/kg), and TCcp (500, 250, and 125 mg/kg) treatments. Groups were designated based on the body weight range (34.80–39.40 g) and the calf thickness range (3.09~3.35 mm) of mice.

Different doses of TCcp such as 500, 250, and 125 mg/kg were orally supplied (in a volume of 10 mL/kg) once daily using a Zonde needle attached to 1 mL syringe for 24 days starting from two weeks before DEXA treatment. Oxymetholone (50 mg/kg) dissolved in distilled water was orally administered as reported previously [33,34,35,43]. The dosages of TCcp (500, 250, and 125 mg/kg) were decided based on previously reported in vivo bioavailability and efficacy studies using tart cherries [43]; and were classed as high, middle, and low dose groups during this study. Catabolic muscular atrophy was induced in mice by subcutaneous DEXA (1 mg/kg, Sigma-Aldrich, St. Louise, MO, USA) treatment around the cervical dorsal back skins, once daily for ten days [16,24]. The GLU and intact vehicle control mice were provided with an equal volume of distilled water instead of TCcp or oxymetholone. Furthermore, an equal volume of saline was subcutaneously administered to intact vehicle control, instead of DEXA.

### 2.3. Measurement of Body Weights

The body weights were recorded one day before treatment, on the day of treatment, and 1, 7, 14, 19, 23, and 24 days after administration of the test materials with an electronic balance (Precisa Gravimetrics AG, Dietikon, Switzerland). Body weight gains were calculated during 14 days of test material pre-administration, 10 days of DEXA treatment, and 24 days of test material administration, to reduce individual differences.

### 2.4. Measurement of Gastrocnemius Muscle and Calf Thickness

The left hind calf thickness was recoded one day before treatment, on the day of treatment, and 1, 7, 14, 19, 23, and 24 days after test material administration using a digital caliper (Mitutoyo, Tokyo, Japan) [35]. In order to reduce the differences from the surrounding tissues, the left hind limb gastrocnemius muscle thickness was calculated after muscular exposure at sacrifice. Changes in calf thickness during the 14 days pre-administration period, 10 days DEXA treatment, and 24 days total period, were determined to limit individual variations.

### 2.5. Measurement of Calf Muscle Strength

One hour after the 24th administration of TCcp, oxymetholone, and vehicle, the muscle strength of calf in each mouse was calculated as tensile strength using an automated testing machine (Japan Instrumentation System Co., Tokyo, Japan) as demonstrated by Kim et al. [43].

### 2.6. Measurement of Gastrocnemius Muscle Weight

After careful separation of gastrocnemius muscles from the fibula and tibia, weights of individually induced gastrocnemius muscles were recorded in grams (absolute wet weights) using an electronic balance. To limit the differences from individual body weights, the percentage body weights (relative body weights) were also calculated.

### 2.7. Serum Biochemical Analysis

Serum biochemical analyses were performed as demonstrated in our previous study [43]. Approximately 200 μL of whole blood sample was drawn from the posterior vena cava using a 26-gauge needle attached syringe under 2–3% isoflurane (Hana Pharm., Hwaseong, Korea) inhalation anesthesia. Blood samples were collected using a separation tube and centrifuged at 1000× *g* for 10 min. Separated serum samples were stored at −150 °C in an ultra-deep freezer (Sanyo, Tokyo, Japan) till further analysis. Serum creatinine, creatine kinase (CK) and lactate dehydrogenase (LDH) levels were measured using auto analyzer (Fuji Medical System Co., Ltd., Tokyo, Japan). Animals were sacrificed by cervical dislocation.

### 2.8. Antioxidant Analysis

Separated gastrocnemius muscles were weighed, homogenized in ice-cold 0.01 M Tris-HCl buffer (pH 7.4), and centrifuged at 12,000× *g* for 15 min as described by Del Rio et al. [44]. Total protein content was measured by following the previously reported method [45] and bovine serum albumin (Invitrogen, Carlsbad, CA, USA) was used as an internal standard.

The gastrocnemius muscles lipid peroxidation concentrations were determined by estimating MDA via thiobarbituric acid test at 525 nm absorbance using a UV/Vis spectrophotometer (Mecasys, Daejeon, Korea). The results were presented as nM of MDA/g tissue [46]. Reactive oxygen species (ROS) level analysis was performed using 2′,7′-dichlorofluorescein diacetate (DCFDA) fluorescent dye as a probe and fluorescence density was measured at 490/520 nm according to the manufacturer’s guidelines of Cellular Reactive Oxygen Species Detection Assay Kit (Abcam, Cambridge, MN, USA). The measured values of optical density [a relative fluorescence unit (RFU)] were corrected by the protein concentrations of samples and were expressed as RFU/μg protein [47]. Glutathione (GSH) content was measured at 412 nm absorbance of 2-nitrobenzoic acid (Sigma-Aldrich, St. Louise, MO, USA) and the results were expressed as mg/g tissue [48]. Decomposition of H_2_O_2_ in the presence of catalase (CAT) was measured at 240 nm [49]. Superoxide dismutase (SOD) estimation was based on the generation of superoxide radicals produced by xanthine and xanthine oxidase, which react with nitrotetrazolium blue to form formazan dye. SOD activity was measured at 560 nm and expressed as U/mg protein.

### 2.9. qRT-PCR

Quality and concentration of TRIzol (Invitrogen, Carlsbad, CA, USA) extracted RNA was determined using a quantitative real-time PCR System (Bio-Rad, Hercules, CA, USA). Recombinant DNase I (DNA-free; Ambion, Austin, TX, USA) was used to remove DNA traces. RNA was reversed-transcribed using a high-capacity cDNA reverse transcription kit (Applied Biosystems, Foster City, CA, USA), and 18 s ribosomal RNA was used as an internal control. Primer sequences are provided in Table 1.

### 2.10. Histopathology

Histological analyses were performed as demonstrated in our previous study [43]. Regions containing collagen fibers (%/mm^2^ of muscle bundles) and mean muscle fiber diameters (µm/fiber) in gastrocnemius muscle samples were histologically assessed.

### 2.11. Immunohistochemistry

Deparaffinization of sectioned histological paraffin and antigen (epitope) retrieval was performed by pretreating in citrate buffer [43]. The antibodies for caspase-3, cleaved poly (ADP-ribose) polymerase (PARP), nitrotyrosine, 4-hydroxynonenal (4-HNE), inducible nitric oxide synthase (iNOS), and myostatin used for immunostaining by avidin-biotin complex (ABC) methods are listed in Table 2. The mean numbers of myostatin-immunoreactive fibers dispersed on each mm^2^ muscle bundles and caspase-3, PARP, nitrotyrosine, 4-HNE, and iNOS activity were calculated by observing the restricted view fields in computer monitor using automated image analysis process (DMI, Daegu, Korea) attached to light microscopy [43]. If the cells or muscle fibers occupied more than 20% of immunoreactivity, the density of each antiserum for caspase-3, PARP, nitrotyrosine, 4-HNE, iNOS and myostatin as compared with intact muscles was regarded as positive.

### 2.12. Statistical Analyses

Multiple comparison tests were performed to compare different dose groups. Variance homogeneity was estimated by Levene’s test. If no significant result was observed, then the data were analyzed by one-way analysis of variance (ANOVA) followed by the least-significant differences multi-comparison (LSD) test. If a significant deviation using the Levene’s test was reported, Kruskal–Wallis H test was executed. If a significant deviation using a Kruskal–Wallis H test was observed, the Mann–Whitney U test was performed to calculate the significantly different pairs in group comparison. SPSS ver. 14 (SPSS Inc., Chicago, IL, USA) was used to conduct the statistical analyses. Furthermore, the percent changes in GLU and intact vehicle control mice were measured to observe the induced severity of catabolic muscular atrophy. The percent variations compared with the test material treated and GLU control mice were also determined to evaluate the effectiveness of the tested materials. The results were considered statistically significant at *p* < 0.01 or *p* < 0.05.

## 3. Results

### 3.1. Body Weight Changes

During 10 days of DEXA treatment and after 24 days of total experimental period, significant (*p* < 0.01) decreases in body weights of GLU control mice were observed as compared with the intact control mice. However, oxymetholone (50 mg/kg) and TCcp (500 and 250 mg/kg) treatments from day 5 of initial DEXA treatment to sacrifice significantly (*p* < 0.01; *p* < 0.05) inhibited the decrease in body weights. A significant (*p* < 0.01) increase in body weight changes in mice treated with test material compared to that treatment with the GLU control was observed after 10 and 24 days of DEXA treatment. Test materials did not show any changes in body weights compared with the intact vehicle or GLU control mice during the 14 days pretreatment period. TCcp (500 and 250 mg/kg) dose-dependently inhibited the GLU-induced decreases in body weights and gains, but no significant changes were demonstrated by TCcp (125 mg/kg) compared with that of GLU control (Figure 1).

### 3.2. Changes in Calf Thickness

Nineteen days after the first treatment, significant (*p* < 0.01) decreases in calf thicknesses in GLU control mice were observed as compared to the intact control. Significant decreases (*p* < 0.01) in calf thickness after day 10 and day 24 of DEXA treatment were observed in GLU control mice compared with the intact vehicle control. Conversely, the decreases in calf thickness were significantly (*p* < 0.01) and dose-dependently inhibited by TCcp (500 and 250 mg/kg) treatments after day 5 of the first DEXA treatment. Significant (*p* < 0.01) increases in calf thickness during 10 days of DEXA treatment and 24 days of test material administration were observed in the experimental groups compared with the intact vehicle control. In the present study, calf thickness of mice treated with oxymetholone after day 5 of the first DEXA treatment showed a significant (*p* < 0.01) increase as compared with the GLU control. TCcp (125 mg/kg) administered groups did not show any significant change in the calf thickness as compared with GLU control mice (Figure 2 and Figure 3).

### 3.3. Effects on Gastrocnemius Muscle Thickness

GLU control mice showed significant (*p* < 0.01) decreases in gastrocnemius muscle thickness after muscular exposure by removing skin as compared to the intact vehicle control. Nevertheless, mice treated with all three dosages of TCcp and oxymetholone showed significant (*p* < 0.01) increases in gastrocnemius muscle thickness compared to the GLU control. TCcp (500 and 250 mg/kg) dose-dependently inhibited the GLU-induced decreases in gastrocnemius muscle thickness; however, no significant variations in gastrocnemius muscle thickness were demonstrated by TCcp (125 mg/kg) as compared to the GLU control (Figure 4).

### 3.4. Changes in Weight of Gastrocnemius Muscle Mass

The significant (*p* < 0.01) decreases in the absolute wet-weight and relative weights of gastrocnemius muscle mass were observed in GLU control mice as compared to the intact vehicle control. Conversely, mice treated with TCcp (500 and 250 mg/kg) and oxymetholone exhibited significant (*p* < 0.01) increases in the weight of gastrocnemius muscle mass as compared with the GLU control. TCcp (500 and 250 mg/kg) dose-dependently inhibited the GLU-induced decreases in gastrocnemius muscle mass weight, but no significant changes were demonstrated by TCcp (125 mg/kg) treated mice as compared with the GLU control (Figure 5).

### 3.5. Changes in Calf Muscle Strength

A significant (*p* < 0.01) decrease in the tensile strength of calf muscles was observed in GLU control mice as compared with intact vehicle control. Whereas, TCcp (500 and 250 mg/kg) and oxymetholone revealed significant (*p* < 0.01) increases in calf muscle strength compared with the GLU control. TCcp (500 and 250 mg/kg) dose-dependently inhibited the GLU-induced decreases in calf muscle strength; whereas no significant changes were demonstrated by TCcp (125 mg/kg) treated mice as compared with the GLU control (Figure 6).

### 3.6. Changes in Serum Biochemistry

The significant (*p* < 0.01) increases in serum creatinine and CK levels, and decreases in serum LDH levels were observed in GLU control mice as compared with the intact vehicle control. Mice treated with TCcp (500 and 250 mg/kg) and oxymetholone revealed significant (*p* < 0.01) increase in serum LDH levels and significant (*p* < 0.01 or *p* < 0.05) decreases in serum creatinine and CK levels compared with the GLU control. TCcp (500 and 250 mg/kg) dose-dependently inhibited the GLU-induced increases in serum CK and creatinine levels and decreases in serum LDH levels. However, no significant effects on serum creatinine, CK, and LDH levels were detected in TCcp (125 mg/kg) treated mice as compared to the GLU control (Table 3).

### 3.7. Changes in Antioxidant Properties

#### 3.7.1. Changes in Muscle MDA Levels and ROS Content

GLU control mice showed significant (*p* < 0.01) increases in MDA levels, ROS content, and muscular lipid peroxidation compared with intact vehicle control. However, TCcp (500 and 250 mg/kg) treated mice significantly (*p* < 0.01; *p* < 0.05) and dose-dependently decreased the elevated MDA and ROS levels. Oxymetholone-treated mice showed significantly (*p* < 0.01) decreased gastrocnemius muscle lipid peroxidation compared with that of GLU control mice; however, TCcp (125 mg/kg) had no significant effect on muscular lipid peroxidation and ROS content compared with the GLU treatment (Table 4).

#### 3.7.2. Changes in Muscle GSH Content and Enzyme Activities

GLU control groups displayed significant (*p* < 0.01) decreases in muscular endogenous antioxidants, antioxidative enzymes, GSH content, and CAT and SOD activity compared with the intact control. However, the 24 days oral treatment with TCcp (500 and 250 mg/kg) and oxymetholone significantly (*p* < 0.01; *p* < 0.05) inhibited the decreases in muscular GSH content, and CAT and SOD activity. TCcp (500 and 250 mg/kg) dose-dependently increased the gastrocnemius muscle GSH content, however, no significant changes in muscular GSH content were demonstrated by TCcp (125 mg/kg) compared with the GLU control (Table 4).

### 3.8. Changes in Gastrocnemius Muscle Protein mRNA Expressions (qRT-PCR Analysis)

Variations in the mRNA expression of sirtuin 1 (SIRT1), muscle RING-finger protein-1 (MuRF1), atrogin-1, and myostatin: GLU control mice showed a significant (*p* < 0.01) increase in mRNA expression of the gastrocnemius muscular SIRT1, MuRF1, atrogin-1, and myostatin, involved in protein degradation, compared with the intact vehicle control. However, increases in muscular MuRF1, atrogin-1, SIRT1, and myostatin mRNA expression were significantly (*p* < 0.01) and dose-dependently decreased by treatment with TCcp (500 and 250 mg/kg). In addition, oxymetholone treatment showed significantly decreased (*p* < 0.01) mRNA expressions of gastrocnemius muscle MuRF1, atrogin-1, SIRT1, and myostatin compared with the GLU control treatment. Treatment with TCcp (125 mg/kg) showed no significant changes in the mRNA expressions as compared with the GLU (Table 5).

GLU control groups showed significant (*p* < 0.01) decreases in the mRNA expression of gastrocnemius muscular transient receptor potential cation channel subfamily V member 4 (TRPV4), phosphatidylinositol 3-kinase (PI3K), adenosine A1 receptor (A1R), and serine-threonine protein kinase (Akt1)-activation compared with the intact control. Nevertheless, TCcp (500 and 250 mg/kg) and oxymetholone treatments revealed significantly (*p* < 0.01; *p* < 0.05) increased muscular PI3K, Akt1, A1R, and TRPV4 mRNA expressions compared with the GLU control mice. TCcp (500 and 250 mg/kg) dose-dependently inhibited the GLU-induced decrease in muscular PI3K, Akt1, A1R, and TRPV4 mRNA expression. However, no significant changes in muscular PI3K, Akt1, A1R, and TRPV4 mRNA expression were demonstrated by TCcp (125 mg/kg) treatment compared with the GLU control (Table 5).

### 3.9. Changes in Gastrocnemius Muscle Histopathology

The DEXA treatment in GLU control mice induced classic and marked changes in the catabolic muscular atrophic mice, including diminished micro focal fibrosis, vacuolation, and muscle fibers in muscle bundles. Accordingly, GLU control mice showed a significant (*p* < 0.01) increase in the percentage of collagen fiber-occupied areas in muscle bundles and decrease in the mean diameter of muscle fibers compared with the intact vehicle control. Whereas TCcp (500 and 250 mg/kg) treated groups dramatically and significantly (*p* < 0.01; *p* < 0.05) decreased the catabolic atrophic changes induced by DEXA treatment in gastrocnemius muscle in a dose-dependent manner. Furthermore, a significant (*p* < 0.01) inhibition in the muscular atrophic changes was observed in oxymetholone-treated mice compared with the GLU control. However, TCcp (125 mg/kg) did not show any significant changes in percentages of collagen fiber-occupied regions in muscle bundles and the mean muscle fiber diameters compared with the GLU control (Table 6, Figure 7).

### 3.10. Changes in Immunohistochemistry of Gastrocnemius Muscle

GLU control mice significantly (*p* < 0.01) increased the apoptotic markers (PARP-immunoreactive fibers, caspase-3-immunoreactive fibers), a lipid peroxidation marker (4-HNE-immunoreactive fibers), an iNOS-related oxidative stress marker (nitrotyrosine-immunoreactive fibers), a potent negative regulator of muscle growth (myostatin-immunoreactive fibers in the gastrocnemius muscle bundles), and an oxidative stress marker (iNOS-immunoreactive fibers). TCcp (500 and 250 mg/kg) dose-dependently and significantly (*p* < 0.01; *p* < 0.05) normalized the GLU-induced variations in the gene expressions. Furthermore, oxymetholone significantly (*p* < 0.01) decreased the numbers of PARP, caspase-3, 4-HNE, nitrotyrosine, myostatin, and iNOS positive muscle fibers. Conversely, no significant changes in muscle fiber PARP, caspase-3, 4-HNE, nitrotyrosine, myostatin, and iNOS immunoreactivity were demonstrated by TCcp (125 mg/kg) compared with the GLU control (Table 6, Figure 8, Figure 9 and Figure 10).

## 4. Discussion

Reduction in muscle tension is a leading cause of atrophy, which increases the protein degradation and reduced protein synthesis [49]. Oxidative stress induces muscle atrophy during disuse or in muscle catabolic cachexia [50]. In addition, the apoptosis of muscle fibers has been related with the early phase of muscular atrophy [51]. The properties of myopathy include insulin resistance, muscle weakness and atrophy, oxidative stress, and mitochondrial dysfunction. Steroid-induced myopathy is symmetric and proximal which may include both lower and upper extremities. It is commonly linked with the use of fluorinated steroids, such as betamethasone, triamcinolone, and GLU, but could also be initiated by non-fluorinated steroids, such as prednisolone and hydrocortisone [52]. Histological changes may include type II-specific atrophy of muscle fibers, loss of myosin filaments in sarcomeres, necrosis, and the preservation of thin filaments and Z-bands [53]. The present study explored the possible beneficial effects of TCcp on the preservation of skeletal muscles in GLU-induced catabolic muscular atrophy in mice.

In this study, a 24 day-experimental period showed normal body weights for all mice in the intact vehicle-control group. GLU treatment decreased the body weight gains which could be due to the cachexia induced by potent catabolic effects of GLU [54]. The body weight increases noted in mice treated with TCcp (500 and 250 mg/kg) may be related to the antioxidative and anti-inflammatory effects of TCcp [39,50]. Generally, enhanced immune system of animals leads to good growth patterns [55]. Oxymetholone, a typical 17α-alkylated anabolic-androgenic steroid, may inhibit the GLU-induced catabolic cachexia-related decreases in body weights due to its strong anabolic effects [56]. High doses of GLU can cause catabolic muscular atrophy which is described by the degradation and reduction of proteins, production of muscle strength, decreased fiber diameter, and resistance to fatigue [23,28,43]. In this study, decreased calf thickness was noted after five days of initial treatment with DEXA. Furthermore, decreased muscle strength of calf weight at sacrifice, and gastrocnemius muscle thickness was observed which could be due to the catabolic muscular atrophy. Inhibition of GLU-induced gastrocnemius muscle thickness, calf muscle thickness and strength, and decreases in weights after oral administration of TCcp (500 and 250 mg/kg), and oxymetholone (50 mg/kg) indicate that TCcp enhanced the GLU-stimulated atrophic changes in calf muscle.

A naturally occurring nitrogenous organic acid, creatinine, is an energy supplier to all body cells, particularly muscle. It is synthesized in the kidney and liver and accumulates in muscle, especially skeletal muscle, through a specific active transport from plasma [57]. Creatinine rapidly diffuses from muscle into urine and plasma with no reuptake into muscle and is not substantially metabolized [57]. Therefore, under state-state condition, creatinine excretion equals creatinine production [57]. Therefore, levels of plasma creatinine could be adjusted to obtain a useful serum biochemical marker to evaluate the amounts, activity, and damage of skeletal muscle [43]. Significant increases in serum creatinine levels were detected, along with GLU-induced changes in catabolic muscle atrophy, comparable to those reported in previous studies [15,43]. However, oral administration of TCcp (500 and 250 mg/kg) dose-dependently and significantly inhibited the increase in serum creatinine levels induced by GLU. This suggests that TCcp has dose-dependent and favorable effects on muscle preservation from GLU-induced muscular atrophy. In the current study, no significant variations in serum creatinine levels were observed with TCcp (125 mg/kg) compared to those observed in GLU control mice.

The medically significant LDH is found widely throughout the body tissues, including CK, heat muscle, and blood cells expressed in various tissues. CK catalyzes the conversion of creatine, which requires adenosine. They could be used as markers of ordinary diseases and injuries, particularly muscle injury as they are discharged during tissue injury. Plasma levels of LDH and CK are used as markers of muscle injury [43], and are markedly elevated in disused muscular atrophic animals [58]. In animals with catabolic muscular atrophy induced by DEXA treatment, marked increases in serum CK levels were observed; however, serum LDH levels decreased due to reduced physiological activities such as contraction of skeletal muscle fiber [59,60]. Significant decreases in serum LDH levels, reduced muscular activity, increases in serum CK levels and muscular damages were observed in GLU control. Nevertheless, dose-dependent and significant decreases in serum CK levels and increases in serum LDH levels were detected in TCcp (500 and 250 mg/kg) treatments, suggesting that TCcp (250 and 500 mg/kg) has promising and powerful effects on muscle preservation.

The oxidative stress due to lipid peroxidation induces muscle atrophy in both muscle cachexia and disused muscles [46]. GSH, an endogenous antioxidant, prevents tissue damage by retaining the ROS at low levels and acts as a defensive antioxidant factors in tissues [61]. SOD contributes to enzymatic defense systems and CAT catalyzes the conversion of H_2_O_2_ to H_2_O [62]. Therefore, inhibition of increased ROS and lipid peroxidation, with increased GSH content, and enzymatic activity in the damaged muscular tissue are of lesser significance for the protection against muscular atrophic changes [63]. The 4-HNE, a lipid peroxidation marker, is produced in cells via lipid peroxidation. It causes numerous diseases, including neurodegenerative diseases, chronic inflammation, different types of cancer, atherogenesis, diabetes, and adult respiratory distress syndrome [64]. Nitrotyrosine, commonly found in pathological conditions, is used as a marker of reactive nitrogen species-induced iNOS-dependent nitrative stress [65]. Furthermore, the destruction of antioxidant defense systems in muscle tissues is involved in GLU-induced catabolic muscular atrophy [43,46].

In the present study, TCcp, dose-dependently protected the gastrocnemius muscle atrophy against GLU-induced oxidative stress (ROS formation), decreased GSH contents and enzymatic (CAT and SOD) activity, and increased lipid peroxidation, 4-HNE-immunolabeled muscle fibers, and nitrotyrosine. This suggests that TCcp (250 and 500 mg/kg) may protect muscle through its potent antioxidative activities, at least partially, under the conditions of the present study. Oxymetholone showed promising antioxidant activity against the GLU-induced inhibition of antioxidant defense mechanisms, which agrees with the previous reports of anabolic steroids [66] and GLU-induced catabolic muscular atrophic changes in mouse [43].

Muscle fiber apoptosis occurs during the early phase of muscular atrophy [47], and PARP and caspase-3 are key drivers of this process [67]. Increase in the amount of PARP and caspase-3 immunoreactive muscle fibers in muscle bundles suggests apoptosis and associated injuries [68]. Moreover, DEXA treatment induces noticeable muscular fiber apoptosis [23,43]. Hence, dose-dependent inhibition of PARP and caspase-3 immunoreactivity in DEXA-treated gastrocnemius muscle bundles following treatment with TCcp (500 and 250 mg/kg) indicates that these treatments can preserve muscle mass by inhibiting GLU-induced muscle fiber apoptosis. However, no significant changes in PARP and caspase-3 immunoreactivity in muscle fiber were observed in mice treated with TCcp (125 mg/kg) compared with GLU control treatment. TCcp (500 mg/kg) presented a weaker inhibitory effect on the GLU-induced increases of PARP and caspase-3 immunoreactive compared with oxymetholone (50 mg/kg). Muscle mass and structure are estimated by the balance between protein degradation and synthesis [4]. In the protein degradation pathway, muscle wasting is mainly linked with ATP–ubiquitin-dependent proteolysis [9], and three enzymes such as E1 (ubiquitin-activating), E2 (ubiquitin-conjugating), and E3 (ubiquitin ligase) are engaged in ubiquitous cascades during this process.

Latest studies have reported that muscle-specific ubiquitin ligases (E3), such as MuRF1 and atrogin-1, play important role in muscular atrophy [69]. Previous studies have shown increased expression of atrogin-1 and MuRF1 in atrophic skeletal muscles, and mice deficient in these genes were resistant to muscle atrophy [5,70]. Furthermore, significant increases in atrogin-1 and MuRF1 mRNA expressions have been noticed in GLU-induced catabolic atrophy [16,28,43]. In our study, GLU control group showed a significant increase in MuRF1 and atrogin-1 mRNA expressions in gastrocnemius muscle compared with the intact vehicle control; however, TCcp (500 and 250 mg/kg) treatments dose-dependently inhibited these increases, but to a lesser degree than by oxymetholone (50 mg/kg). The results indicate that TCcp potently protected muscle by down regulating the expression of atrogin-1 and MuRF1, which are related with the degradation of muscle protein, at dosage of 250 and 500 mg/kg, but to a lesser extent than observed with oxymetholone.

The insulin-like growth factor 1 (IGF-1)-PI3K-Akt signaling pathway plays an important role in the initiation of protein synthesis [6]. PI3K, activated by insulin or IGF, in turn stimulates Akt, a serine/threonine kinase, mammalian target of rapamycin complex (mTOR), and phosphorylates GSK-3β, thereby inducing hypertrophy [71]. Marked down regulated mRNA expressions of Akt1 and PI3K were observed in GLU control, indicating changes in catabolic muscular atrophy, which is similar with the findings of a previous study [43]. However, TCcp (500 and 250 mg/kg) dose-dependently up-regulated the PI3K and Akt1 mRNA expressions as compared with that in the GLU control, but lesser degree than observed with oxymetholone. This indicates that TCcp can initiate protein synthesis is muscle and resists GLU-induced catabolic muscular atrophy, but to a lesser degree than observed with oxymetholone.

Adenosine modulates numerous physiological functions of the cardiovascular system in a variety of tissues including skeletal muscle [72]. It regulates blood flow to skeletal muscle [73], and synergistically affects insulin-stimulated glucose uptake and contraction in skeletal muscle [74]. Most of the physiological effects of adenosine are regulated via specific adenosine receptors [75], such as A1R, a cytoprotective of skeletal muscle [76]. TRPV4, a member of TRP channel superfamily plays a mechanosensory or osmosensory function in many musculoskeletal tissues and inhibits bone loss or muscular atrophy [77]. Subcutaneous DEXA treatment significantly decreased the mRNA expression of TRPV4 and A1R in gastrocnemius muscle, and catabolic muscular atrophy-related proteolysis, like the previous study [43]. TCcp (500 and 250 mg/kg) dose-dependently up-regulated A1R and TRPV4 expressions, which are involved in muscle growth, compared with the intact control. However, this was not observed with TCcp (125 mg/kg) and occurred to a lesser extent than with oxymetholone. This provides direct evidence that TCcp (500 and 250 mg/kg) resists GLU-induced catabolic muscle atrophy and can enhance muscle growth, but to a lesser extent than oxymetholone. Similar results were observed with TCcp (500 mg/kg) and oxymetholone.

Myostatin inhibits muscle growth and differentiation during myogenesis [9,16]. It is mainly produced in muscle cells of skeletal, flows in the blood stream, and acts on muscle by binding activin type II receptor on the cell surface [78]. The Sirtuin family of proteins possesses NAD^+^-dependent and/or ADP-ribosyltransferase and deacetylase activity. Seven mammalian Sirtuins (SIRT1–7) are located differentially within the cell and have numerous roles [79]. Among those, SIRT1 is the most extensively reported, and it controls various biological activities ranging from cell differentiation, metabolism, proliferation, and apoptosis [80]. During catabolic muscular atrophy, DEXA treatments have reported decreased mRNA expressions of SIRT1 and myostatin and muscular mass [16,25,43]. However, the TCcp (500 and 250 mg/kg) treatments dose-dependently inhibited increased mRNA expressions of SIRT1 and myostatin; this indicates a strong muscle protective effect of TCcp by down regulating SIRT1 and myostatin expressions.

GLU-induced catabolic muscular atrophic changes show characteristic histopathological changes, including diminished micro vacuolation and collagen depositions, muscle fiber diameters, protein degradation, and fibrosis [15,21,43]; similar results were also noticed during this study. The histopathological inhibition of muscular atrophic variations following treatment with TCcp (500 and 250 mg/kg) indicates that TCcp can preserve GLU-induced catabolic muscular atrophy. No significant changes in the percentage collagen fiber-occupied regions or the mean muscle fiber diameters in muscle bundles were observed in TCcp (125 mg/kg) treatment compared with DEXA treatment.

TCcp (500 mg/kg) had a weak inhibitory effect on GLU-induced atrophic changes and histopathological muscle fibrosis compared with the oxymetholone. During this study, TCcp (500 mg/kg) had a weak inhibitory effect on the GLU-induced decreases in body weight, calf thickness, muscle thickness, gastrocnemius muscle weight, calf muscle strength, decreased serum LDH levels, muscular GSH levels, SOD and CAT activity, muscle PI3K, Akt1, A1R, and TRPV4 mRNA expression levels, decreased mean muscle fiber diameter and increases in serum creatinine and CK levels, muscular lipid peroxidation and ROS content, mRNA expressions of muscular SIRT1, MuRF1, atrogin-1, and myostatin, increased percentage of collagen fiber-occupied regions in muscle bundles, increases in PARP, caspase-3, 4-HNE, nitrotyrosine, myostatin, and iNOS immunoreactivity in muscle fibers as compared with oxymetholone (50 mg/kg).

De Souza and Hallak [81] showed that long-term, non-therapeutic doses of oxymetholone can increase atrophy and reduce germinal cells in the testicles and also change the serum levels of LH, FSH hormones by increasing free radicals and changing serum testosterone levels [82]. It has been reported that the detrimental side effects of oxymetholone including infertility and enhanced MDA levels in mice [83,84]. Therefore, it is expected that TCcp evaluated in this study could prevent muscle atrophy and muscle loss, at least in patients with muscular atrophy due to side effects of adrenocorticosteroids preparations or inflammatory elderly patients who frequently use adrenocorticosteroids preparations. However, the current study cannot be accurately applied to clinical practice due to an animal experiment using one model, male ICR mice. Hence, the drug efficacy evaluation and clinical trials in other experimental animal models of muscle atrophy such as other sciatic nerve cutting models are necessary.

## 5. Conclusions

The results of this study suggest that TCcp (250 and 500 mg/kg) has a promising effect on the GLU-induced catabolic muscular atrophy, through antioxidative and anti-inflammatory functions, mediated by the modulation of genes involved in the protein degradation (MuRF1, atrogin-1, SIRT1, and myostatin) and synthesis (PI3K, Akt1, TRPV4, and A1R) of muscle. During this study clear muscle atrophy inhibitory effects were recognized. Therefore, it is expected that TCcp at a concentration of 500 mg/kg and 250 mg/kg could improve muscular atrophies originating from various etiologies.

## Figures and Tables

**Figure 1 medicina-57-00485-f001:**
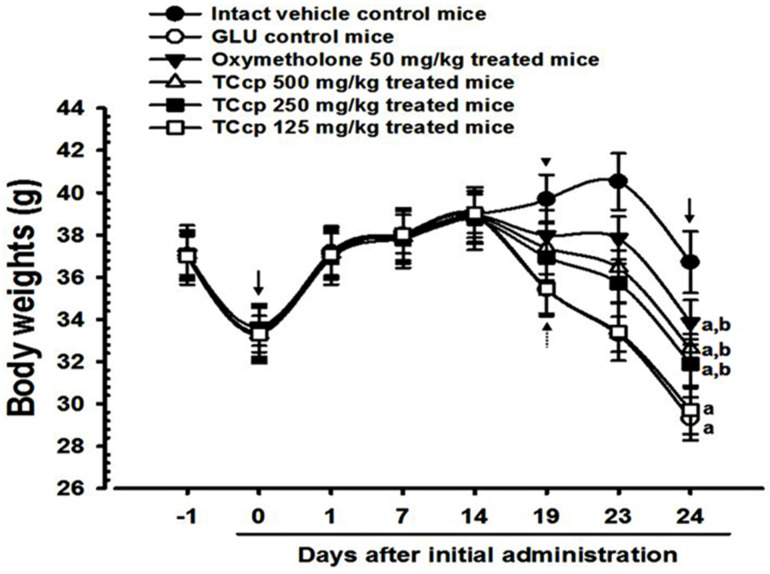
Changes in body weight in GLU-induced muscular atrophic mice. Values are expressed means ± S.D. of eight mice. Dot arrows show significant decreases in body weights in GLU control mice after 5 days of initial DEXA treatment to sacrifice. However, these decreases in body weights were significantly inhibited by treatments with oxymetholone and TCcp (500 and 250 mg/kg) (arrowhead). Day-1 and Day-24 means 1 day before starting the test material administration and at sacrifice, respectively. Day 0 means at start of test material administration (2 weeks before initial DEXA treatment). All animals were overnight fasted before first test material administration and sacrifice (arrows). TCcp = tart cherry concentrated powder. GLU = glucocorticoid. DEXA = dexamethasone. ^a^
*p* < 0.01 as compared with intact control by LSD test. ^b^
*p* < 0.01 as compared with GLU control by LSD test.

**Figure 2 medicina-57-00485-f002:**
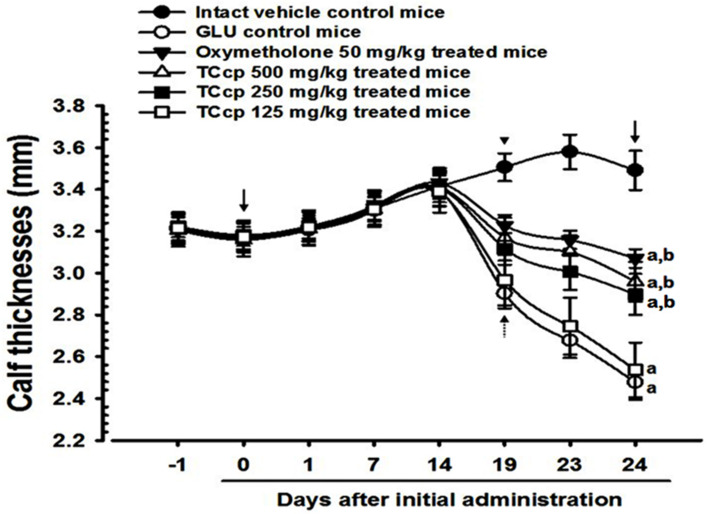
Changes in calf thickness in GLU-induced muscular atrophic mice. Values are expressed means ± S.D. of eight mice. Dot arrows show significant decreases in calf thickness in GLU control mice after 5 days of initial DEXA treatment to sacrifice as compared with the intact control. However, these decreases in calf thicknesses were significantly and dose-dependently inhibited by oxymetholone, TCcp 500 and 250 mg/kg treatments (arrowhead). Day-1 means 1 day before start of the test material administration and day-24 means at sacrifice. Day 0 means at start of test material administration (2 weeks before initial DEXA treatment). All animals were overnight fasted before first test material administration and sacrifice (arrows). TCcp = tart cherry concentrated powder. GLU = glucocorticoid. DEXA = dexamethasone. ^a^
*p* < 0.01 as compared with intact control by LSD test. ^b^
*p* < 0.01 as compared with GLU control by LSD test.

**Figure 3 medicina-57-00485-f003:**
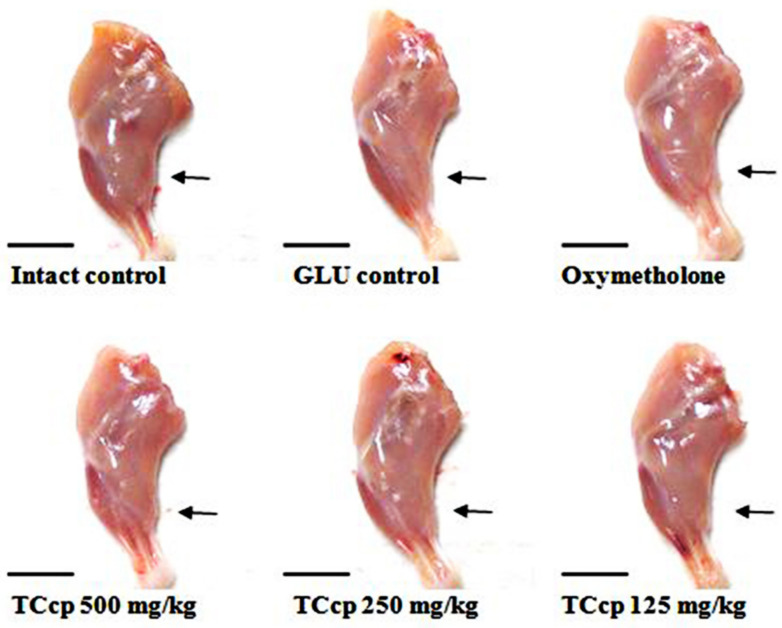
Variations in calf muscle masses in GLU-induced muscular atrophic mice. The thickness of left hind calf was measured at day 24 of test material administration using an electronic digital caliper and shown as mm/mouse levels. TCcp = tart cherry concentrated powder. GLU = glucocorticoid. DEXA = dexamethasone. Scale bars = 9 mm.

**Figure 4 medicina-57-00485-f004:**
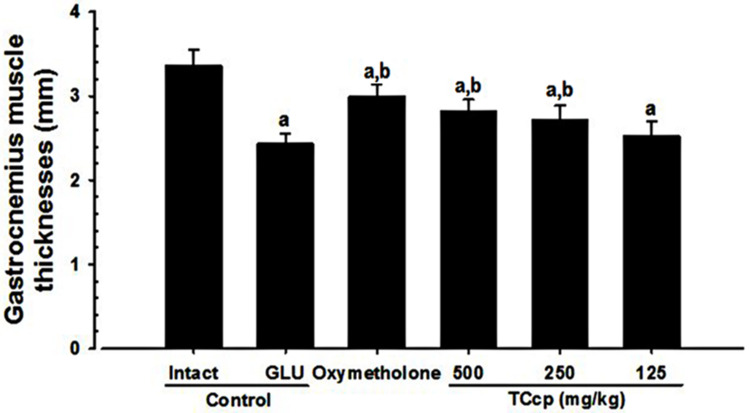
Changes in gastrocnemius muscle thicknesses after muscular exposure in GLU-induced muscular atrophic mice. The thickness of gastrocnemius muscle was measured at day 24 of test material administration using an electronic digital caliper and expressed as mm/mouse levels. Values are expressed means ± S.D. of eight mice. TCcp = tart cherry concentrated powder. GLU = glucocorticoid. DEXA = dexamethasone. Oxymetholone (50 mg/kg) dissolved in distilled water was orally administered. ^a^
*p* < 0.01 as compared with intact control by LSD test. ^b^
*p* < 0.01 as compared with GLU control by LSD test.

**Figure 5 medicina-57-00485-f005:**
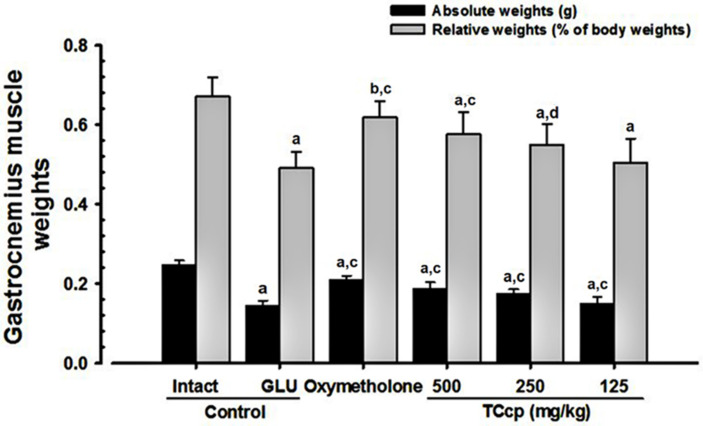
Variations in gastrocnemius muscle weights in GLU-induced muscular atrophic mice.Values are expressed means ± S.D. of eight mice. The gastrocnemius muscle mass weight was measured at day 24 of test material administration and expressed at g levels (absolute wet-weights) using automatic electronic balance. TCcp = tart cherry concentrated powder. GLU = glucocorticoid. DEXA = dexamethasone. Oxymetholone (50 mg/kg) dissolved in distilled water was orally administered. ^a^
*p* < 0.01 and ^b^
*p* < 0.05 as compared with intact control by LSD test. ^c^
*p* < 0.01 and ^d^
*p* < 0.05 as compared with GLU control by LSD test.

**Figure 6 medicina-57-00485-f006:**
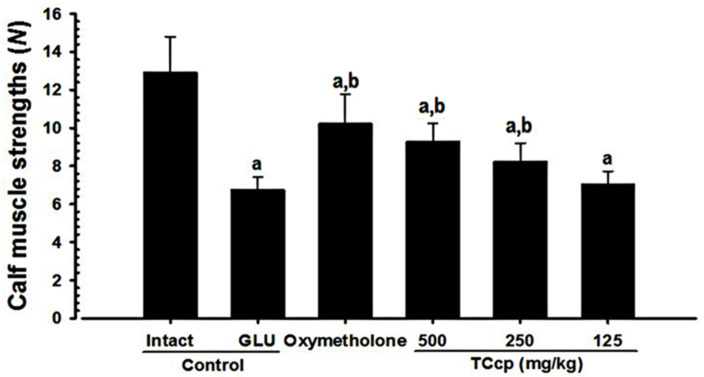
Changes in calf muscle strengths in GLU-induced muscular atrophic mice. Values are expressed means ± S.D. of eight mice. One hour after last (24th) administration of the test materials, the calf muscle strengths of individual mice were measured as tensile strengths using a computerized testing machine and expressed as Newton (*N*). TCcp = tart cherry concentrated powder. GLU = glucocorticoid. DEXA = dexamethasone. *N* = Newton. Oxymetholone (50 mg/kg) dissolved in distilled water was orally administered. ^a^
*p* < 0.01 as compared with intact control by MW test. ^b^
*p* < 0.01 as compared with GLU control by MW test.

**Figure 7 medicina-57-00485-f007:**
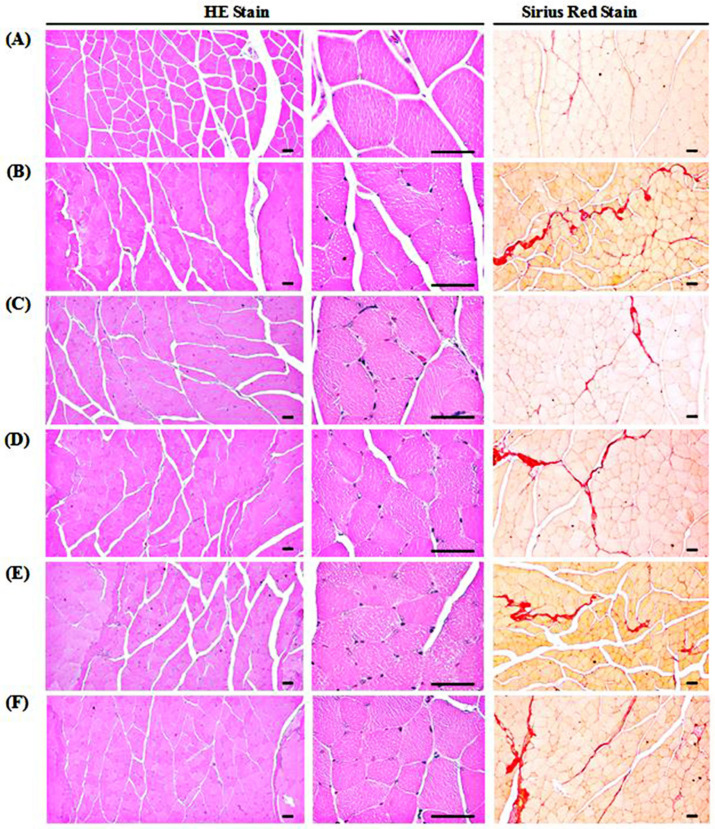
Representative gastrocnemius muscle histology. (**A**) Intact vehicle control (distilled water administered and treated with saline). (**B**) GLU control (distilled water administered and treated with DEXA). (**C**) Oxymetholone (50 mg/kg oxymetholone administered and treated with DEXA as reference). (**D**) TCcp500 (500 mg/kg TCcp administered and treated with DEXA–higher dose treated mice). (**E**) TCcp250 (250 mg/kg TCcp administered and treated with DEXA–middle dose treated mice). (**F**) TCcp125 (125 mg/kg TCcp administered and treated with DEXA–low dose treated mice). TCcp = tart cherry (Fruit of *Prunus cerasus* L., Rosaceae) concentrated powder. GLU = glucocorticoid. DEXA = dexamethasone. Arrows indicated collagen fibers deposited as focal fibrosis. Scale bars = 40 μm.

**Figure 8 medicina-57-00485-f008:**
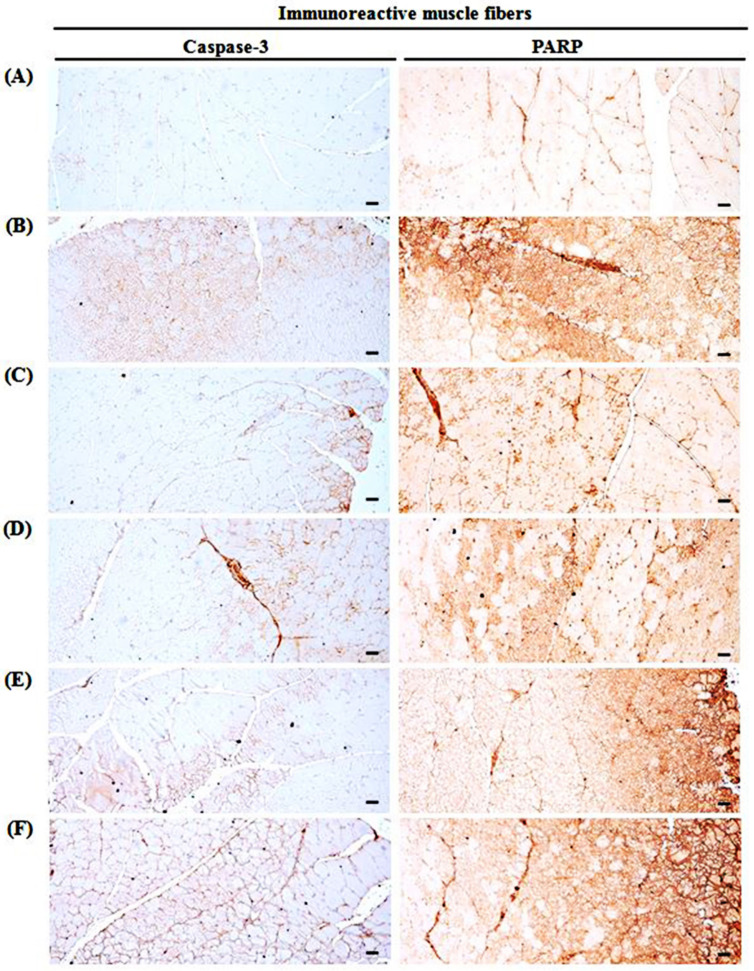
Representative gastrocnemius muscle caspase-3 and PARP-immunoreactivity. (**A**) Intact vehicle control (distilled water administered and treated with saline). (**B**) GLU control (distilled water administered and treated with DEXA). (**C**) Oxymetholone (50 mg/kg oxymetholone administered and treated with DEXA as reference). (**D**) TCcp500 (500 mg/kg TCcp administered and treated with DEXA–higher dose treated mice). (**E**) TCcp250 (250 mg/kg TCcp administered and treated with DEXA–middle dose treated mice). (**F**) TCcp125 (125 mg/kg TCcp administered and treated with DEXA–low dose treated mice). TCcp = tart cherry (Fruit of *Prunus cerasus* L., Rosaceae) concentrated powder. GLU = glucocorticoid. DEXA = dexamethasone. PARP = cleaved poly (ADP-ribose) polymerase. All avidin-biotin complex methods. Scale bars = 40 μm.

**Figure 9 medicina-57-00485-f009:**
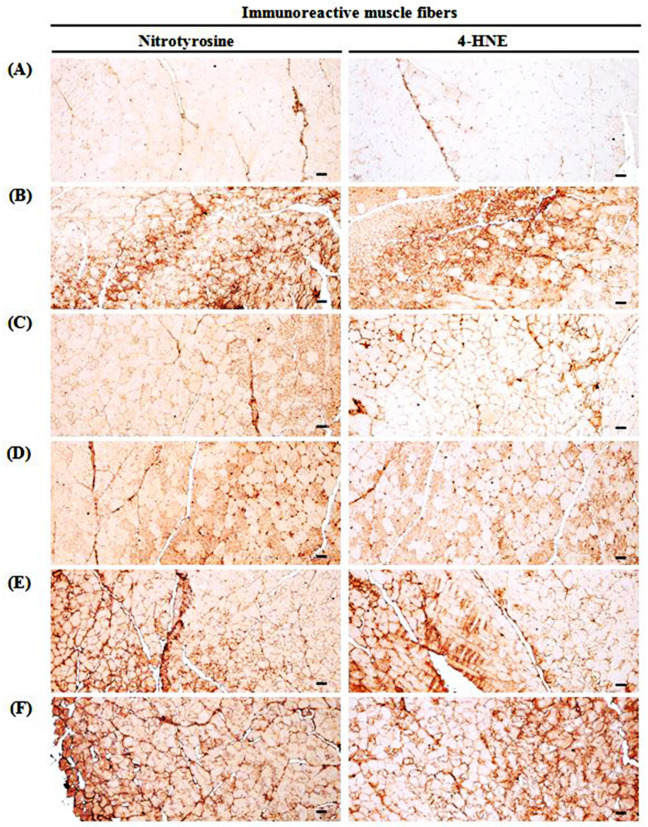
Selective gastrocnemius muscle nitrotyrosine and 4-HNE-immunoreactivities. (**A**) Intact vehicle control (distilled water administered and treated with saline). (**B**) GLU control (distilled water administered and treated with DEXA). (**C**) Oxymetholone (50 mg/kg oxymetholone administered and treated with DEXA as reference). (**D**) TCcp500 (500 mg/kg TCcp administered and treated with DEXA–higher dose treated mice). (**E**) TCcp250 (250 mg/kg TCcp administered and treated with DEXA–middle dose treated mice). (**F**) TCcp125 (125 mg/kg TCcp administered and treated with DEXA–low dose treated mice). TCcp = tart cherry (Fruit of *Prunus cerasus* L., Rosaceae) concentrated powder. GLU = glucocorticoid. DEXA = dexamethasone. 4-HNE = 4-hydroxynonenal. All avidin-biotin complex methods. Scale bars = 40 μm.

**Figure 10 medicina-57-00485-f010:**
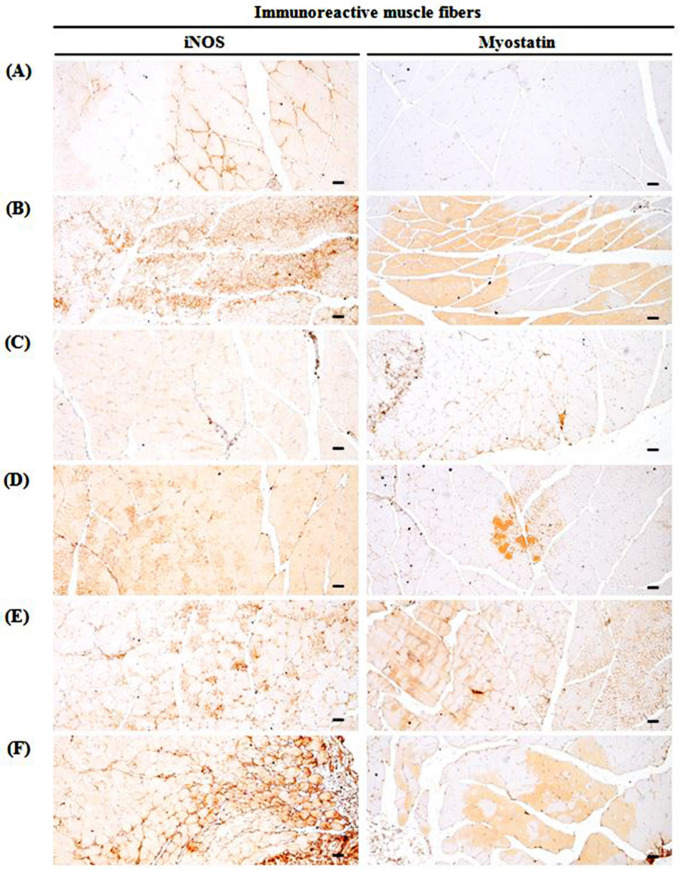
Selective gastrocnemius muscle iNOS and myostatin-immunoreactivity. (**A**) Intact vehicle control (distilled water administered and treated with saline). (**B**) GLU control (distilled water administered and treated with DEXA). (**C**) Oxymetholone (50 mg/kg oxymetholone administered and treated with DEXA as reference). (**D**) TCcp500 (500 mg/kg TCcp administered and treated with DEXA–higher dose treated mice). (**E**) TCcp250 (250 mg/kg TCcp administered and treated with DEXA–middle dose treated mice). (**F**) TCcp125 (125 mg/kg TCcp administered and treated with DEXA–low dose treated mice). TCcp = tart cherry (Fruit of *Prunus cerasus* L., Rosaceae) concentrated powder. GLU = glucocorticoid. DEXA = dexamethasone. iNOS = inducible nitric oxide synthase. All avidin-biotin complex methods. Scale bars = 40 μm.

**Table 1 medicina-57-00485-t001:** Oligonucleotides for qRT-PCR used in this study.

Target	5′–3′	Sequence	Size (bp)	Gene ID
Atrogin-1	Forward Reverse	CAGCTTCGTGAGCGACCT CGGCAGTCGAGAAGTCCAGTC	244	67731
MuRF 1	Forward Reverse	GACAGTCGCATTTCAAAGCA GCCTAGCACTGACCTGGAAG	194	433766
PI3K p85α	Forward Reverse	GCCAGTGGTCATTTGTGTTG ACACAACCAGGGAAGTCCAG	236	18708
Akt 1	Forward Reverse	ATGAACGACGTAGCCATTGTG TTGTAGCCAATAAAGGTGCCAT	116	11651
Adenosine A1R	Forward Reverse	TGTTCCCAGGGCCTTTCAC TAATGGACTGAGACTAGCTTGACTGGTA	155	11539
TRPV4	Forward Reverse	CAGGACCTCTGGAAGAGTGC AAGAGCTAGCCTGGACACCA	165	63873
Myostatin	Forward Reverse	CCTCCACTCCGGGAACTGA AAGAGCCATCACTGCTGTCATC	185	17700
SIRT1	Forward Reverse	TTCACATTGCATGTGTGTGG TGAGGCCCAGTGCTCTAACT	175	93759
18s Ribosomal RNA	Forward Reverse	AGCCTGAGAAACGGCTACC TCCCAAGATCCAACTACGAG	252	19791

qRT-PCR = real-time quantitative reverse transcription polymerase chain reaction. MuRF1 = muscle RING-finger protein-1. PI3k = phosphatidylinositol 3-kinase. A1R = A1 receptor. TRPV4 = transient receptor potential cation cannel subfamily V member 4. SIRT1 = Sirtuin 1.

**Table 2 medicina-57-00485-t002:** Primary antisera and detection kits used in this study.

Antisera or Detection Kits	Code	Source	Dilution
Primary Antisera			
Anti-cleaved caspase-3 (Asp175) polyclonal antibody	9661	Cell Signaling Technology Inc. ^1^	1:400
Anti-cleaved PARP (Asp214) specific antibody	9545	Cell Signaling Technology Inc. ^1^	1:100
Anti-4-Hydroxynonenal polyclonal antibody	Ab46545	Abcam ^2^	1:100
Anti-Nitrotyrosine polyclonal antibody	06-284	Millipore Corporation ^3^	1:200
Anti-nitric oxide synthase2 (N-20) polyclonal antibody	sc-651	Santa Cruz Biotechnology ^4^	1:100
Anti-GDF8/Myoststin antibody	Ab71808	Abcam ^2^	1:50
Detection kits			
Vectastain Elite ABC Kit	PK-6200	Vector Lab. Inc. ^4^	1:50
Peroxidae substrate kit	SK-4100	Vector Lab. Inc. ^4^	1:50

All antiserums were diluted by 0.01 M phosphate buffered saline, in this experiment. ^1^ Danvers, MA, USA. ^2^ Cambridge, UK. ^3^ Billerica, CA, USA. ^4^ Burlingame, CA, USA.

**Table 3 medicina-57-00485-t003:** Changes in the serum biochemistry.

Items (Unit) Groups	Serum Levels		
	Creatinine (mg/dL)	Creatine Kinase (IU/L)	LDH (IU/L)
Controls			
Intact	0.34 ± 0.04	83.75 ± 20.86	665.50 ± 128.94
GLU	0.86 ± 0.13 ^a^	281.63 ± 55.15 ^d^	168.25 ± 50.55 ^d^
Reference			
Oxymetholone	0.44 ± 0.06 ^b,c^	148.00 ± 17.78 ^d,e^	318.50 ± 79.79 ^d,e^
TCcp treated			
500 mg/kg	0.55 ± 0.10 ^a,c^	186.00 ± 15.94 ^d,e^	281.50 ± 25.53 ^d,e^
250 mg/kg	0.64 ± 0.09 ^a,c^	208.88 ± 20.95 ^d,f^	241.13 ± 38.72 ^d,e^
125 mg/kg	0.83 ± 0.10 ^a^	270.25 ± 40.21 ^d^	178.88 ± 61.66 ^d^

Values are expressed means ± S.D. of eight mice. Blood samples were collected at the day of necropsy and serum creatinine, creatine kinase and LDH levels were measured using auto analyzer. TCcp = tart cherry concentrated powder. GLU = glucocorticoid. LDH = lactate dehydrogenase. Oxymetholone was orally administered at 50 mg/kg levels, dissolved in distilled water. ^a^
*p* < 0.01 and ^b^
*p* < 0.05 as compared with intact control by LSD test. ^c^
*p* < 0.01 as compared with GLU control by LSD test. ^d^
*p* < 0.01 as compared with intact control by MW test. ^e^
*p* < 0.01 and ^f^
*p* < 0.05 as compared with GLU control by MW test.

**Table 4 medicina-57-00485-t004:** Changes in gastrocnemius muscle antioxidant defense systems.

Items (Unit) Groups	Fundus Antioxidant Defense Systems				
	Malondialdehyde (nM/mg Protein)	Reactive Oxygen Species (RFU/μg Protein)	Glutathione (nM/mg Protein)	Superoxide Dismutase (nM/mim/mg Protein)	Catalase (U/mg Protein)
Controls					
Intact	1.87 ± 0.75	22.57 ± 10.53	0.66 ± 0.16	35.39 ± 13.08	7.25 ± 2.02
GLU	8.54 ± 1.22 ^a^	70.01 ± 15.85 ^d^	0.17 ± 0.07 ^a^	11.97 ± 2.11 ^d^	2.07 ± 0.56 ^d^
Reference					
Oxymetholone	4.53 ± 0.93 ^a,b^	32.57 ± 11.84 ^f^	0.41 ± 0.12 ^a,b^	22.79 ± 4.32 ^e,f^	3.88 ± 0.63 ^d,f^
TCcp treated					
500 mg/kg	5.19 ± 0.89 ^a,b^	39.40 ± 11.09 ^e,f^	0.31 ± 0.10 ^a,c^	19.36 ± 2.79 ^d,f^	3.56 ± 0.99 ^d,f^
250 mg/kg	6.17 ± 1.25 ^a,b^	46.36 ± 13.36 ^d,g^	0.28 ± 0.06 ^a,c^	17.39 ± 2.19 ^d,f^	3.21 ± 0.92 ^d,f^
125 mg/kg	8.22 ± 1.40 ^a^	66.31 ± 22.46 ^d^	0.18 ± 0.08 ^a^	12.77 ± 2.59 ^d^	2.19 ± 0.59 ^d^

Values are expressed means ± S.D. of eight mice. Separated gastrocnemius muscles were weighed and homogenized in ice-cold 0.01 M Tris-HCl buffer (pH 7.4), and centrifuged at 12,000× *g* for 15 min. The malondialdehyde, reactive oxygen species, glutathione content, superoxide dismutase, and catalase enzyme activities in individual muscles were assessed. TCcp = tart cherry concentrated powder. GLU = glucocorticoid. Oxymetholone was orally administered at 50 mg/kg levels, dissolved in distilled water. ^a^
*p* < 0.01 as compared with intact control by LSD test. ^d^
*p* < 0.01 and ^e^
*p* < 0.05 as compared with intact control by MW test. ^b^
*p* < 0.01 and ^c^
*p* < 0.05 as compared with GLU control by LSD test. ^f^
*p* < 0.01 and ^g^
*p* < 0.05 as compared with GLU control by MW test.

**Table 5 medicina-57-00485-t005:** Changes in gastrocnemius muscle mRNA expressions.

Groups Targets	Controls		Reference	TCcp Treated Mice (mg/kg)		
	Intact	GLU	Oxymetholone	500	250	125
Atrogin-1	0.99 ± 0.07	4.89 ± 0.69 ^c^	2.26 ± 0.55 ^c,e^	3.00 ± 0.74 ^c,e^	3.59 ± 0.81 ^c,e^	4.59 ± 0.64 ^c^
MuRF 1	1.08 ± 0.22	6.29 ± 1.10 ^c^	3.06 ± 0.67 ^c,e^	4.36 ± 1.07 ^c,f^	4.76 ± 0.67 ^c,e^	5.99 ± 1.88 ^c^
PI3K p85α	1.04 ± 0.14	0.59 ± 0.11 ^c^	1.15 ± 0.37 ^e^	0.88 ± 0.10 ^d,e^	0.77 ± 0.12 ^c,f^	0.64 ± 0.14 ^c^
Akt 1	1.02 ± 0.05	0.54 ± 0.06 ^c^	0.87 ± 0.11 ^d,e^	0.74 ± 0.06 ^c,e^	0.67 ± 0.07 ^c,e^	0.57 ± 0.12 ^c^
A1R	1.05 ± 0.14	0.51 ± 0.13 ^a^	0.87 ± 0.07 ^a,b^	0.77 ± 0.08 ^a,b^	0.70 ± 0.10 ^a,b^	0.54 ± 0.17 ^a^
TRPV4	1.11 ± 0.10	0.37 ± 0.09 ^a^	0.66 ± 0.10 ^a,b^	0.61 ± 0.13 ^a,b^	0.55 ± 0.10 ^a,b^	0.40 ± 0.10 ^a^
Myostatin	1.03 ± 0.10	7.02 ± 0.99 ^c^	3.23 ± 0.78 ^c,e^	4.72 ± 1.01 ^c,e^	5.12 ± 0.79 ^c,e^	6.51 ± 1.24 ^c^
SIRT1	1.03 ± 0.17	10.73 ± 3.03 ^c^	3.73 ± 1.11 ^c,e^	4.57 ± 1.19 ^c,e^	6.27 ± 1.35 ^c,e^	10.02 ± 1.57 ^c^

Values are expressed means ± S.D. of eight mice, relative expressions/18 s ribosomal RNA. TCcp = tart cherry concentrated powder. GLU = glucocorticoid. MuRF1 = muscle RING-finger protein-1. PI3k = phosphatidylinositol 3-kinase. A1R = adenosine A1 receptor. TRPV4 = transient receptor potential cation cannel subfamily V member 4. SIRT1 = sirtuin 1. Oxymetholone was orally administered at 50 mg/kg levels, dissolved in distilled water. ^a^
*p* < 0.01 as compared with intact control by LSD test. ^c^
*p* < 0.01 and ^d^
*p* < 0.05 as compared with intact control by MW test, ^b^
*p* < 0.01 as compared with GLU control by LSD test. ^e^
*p* < 0.01 and ^f^
*p* < 0.05 as compared with GLU control by MW test.

**Table 6 medicina-57-00485-t006:** Changes in the gastrocnemius muscle histomorphological analysis.

Groups Index	Controls		Reference	TCcp Treated Mice (mg/kg)		
	Intact	GLU	Oxymetholone	500	250	125
General histomorphometry						
Fiber diameter (μm)	50.88 ± 11.00	23.10 ± 5.10 ^a^	38.38 ± 7.06 ^a,b^	34.85 ± 4.20 ^a,b^	31.67 ± 5.12 ^a,b^	24.94 ± 5.92 ^a^
Collagen (%)	4.23 ± 1.65	32.84 ± 4.89 ^a^	17.69 ± 5.06 ^a,b^	21.56 ± 2.96 ^a,b^	23.82 ± 3.51 ^a,b^	31.32 ± 4.81 ^a^
Immunohistomorphometry (fibers/mm^2^)						
Caspase-3	2.38 ± 2.50	42.13 ± 11.06 ^d^	21.00 ± 4.81 ^d,e^	27.38 ± 3.16 ^d,e^	29.75 ± 5.55 ^d,f^	39.50 ± 5.37 ^d^
PARP	5.13 ± 2.59	76.25 ± 10.95 ^a^	33.25 ± 10.35 ^a,b^	45.88 ± 10.45 ^a,b^	54.88 ± 10.49 ^a,b^	71.25 ± 12.06 ^a^
Nitrotyrosine	5.13 ± 2.64	68.38 ± 12.08 ^a^	32.13 ± 12.73 ^a,b^	46.88 ± 10.99 ^a,b^	51.50 ± 10.97 ^a,b^	66.00 ± 14.77 ^a^
4-HNE	3.63 ± 2.07	76.50 ± 12.17 ^a^	44.13 ± 10.83 ^a,b^	48.75 ± 11.23 ^a,b^	56.75 ± 12.58 ^a,b^	71.50 ± 12.86 ^a^
iNOS	8.00 ± 3.16	52.88 ± 12.19 ^d^	21.75 ± 6.48 ^d,e^	34.00 ± 6.48 ^d,e^	38.25 ± 8.71 ^d,f^	51.00 ± 15.51 ^d^
Myostatin	1.38 ± 1.06	52.75 ± 10.63 ^d^	22.13 ± 4.76 ^d,e^	29.00 ± 8.73 ^d,e^	36.13 ± 11.24 ^d,e^	50.50 ± 11.38 ^d^

Values are expressed means ± S.D. of eight mice. TCcp = tart cherry concentrated powder. GLU = glucocorticoid. PARP = cleaved poly(ADP-ribose) polymerase. 4-HNE = 4-hydroxynonenal. iNOS = inducible nitric oxide synthase. Oxymetholone was orally administered at 50 mg/kg levels, dissolved in distilled water. ^a^
*p* < 0.01 as compared with intact control by LSD test. ^d^
*p* < 0.01 as compared with intact control by MW test. ^b^
*p* < 0.01 as compared with GLU control by LSD test. ^e^
*p* < 0.01 and ^f^
*p* < 0.05 as compared with GLU control by MW test.

## Data Availability

Data supporting reported results can be made available on demand.

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
