# Peer review of "Tart Cherry (Fruit of Prunus cerasus) Concentrated Powder (TCcp) Ameliorates Glucocorticoid-Induced Muscular Atrophy in Mice"

_medicina, 2021, doi:10.3390/medicina57050485_

Round 1

Reviewer 1 Report

The manuscript, “Tart cherry (Fruit of Prunus cerasus) concentrated powder (TCcp) ameliorates glucocorticoid-induced muscular atrophy in mice” investigates the ability of TCcp to prevent dexamethasone-induced muscle wasting and associated effects across a variety of parameters of muscle health. The evidence presented indicates that high doses of TCcp (250-500 mg/kg) is protective against dexamethasone-mediated muscle atrophy and is comparable to their positive control, oxymetholone. Overall, these data are intriguing and convincing; however, there are several issues that need to be addressed. The manuscript requires substantial revision in regards to writing and grammar for proper interpretation of these data. Additionally, several crucial details need to be added or clarified and are specified below.

  • General:
    • There are several abbreviations used throughout the manuscript that are undefined.
    • The experimental design needs clarification throughout, several times it is described in a way that is confusing. For example “during the first 10 days of GLU treatment and 24 days after initial GLU treatment…” (line 210).
    • The abbreviation GLU is used for dexamethasone, but is also referred to as DEXA. It is unclear if the term GLU always refers to dexamethasone or to other glucocorticoids as well. It is important to be clear when referencing specific corticosteroids since glucocorticoid analogs differ greatly in their potency, half-life and specificity for the GR.
    • There is unnecessary redundancy in regards to results and other details throughout.
    • Several references are either out of order or incorrect making it difficult to research certain statements made by the authors, please correct.
    • There are detrimental side effects of oxymetholone that should be mentioned and cited within the manuscript, including infertility and enhanced MDA levels in mice. Future studies should investigate whether there are similar compounds within TCcp and examine whether similar side effects exist with this botanical extract.
    • It would be nice to see some western data to support the mRNA findings as mRNA expression does not always coincide with translation of protein. This would be especially useful when evaluation pathways such as Akt and other proteins whose activity and function depends on posttranslational modifications and less on amount of expression.
    • It appears that only male mice were used for these experiments, but that is not clearly stated. Please clarify and consider including female mice in future studies.
    • Future studies should consider evaluation more than one muscle group and including NMR/MRI to evaluate changes in total lean mass, although when focusing on one the gastrocnemius is a good choice for several reasons.
  • Methods:
    • SPF ICR mice- the abbreviations are not noted. Please explain why you chose this strain for these studies.
    • The catalog number for TCcp needs to be included and the information for the vehicle needs clarification. For example, ‘Pink TCcp powder was dissolved at a concentration of 50mg/mL in distilled water and used as vehicle’ (line 89-90) is quite confusing.
    • It seems that all groups with the exclusion of vehicle are given GLU, but this was not clearly stated and could be confusing to the reader.
    • It is unclear whether the entire triceps surae was measured or the soleus was removed and just the gastrocnemius was assessed for weight and thickness. I would suggest avoiding the use of “calf”.
    • The details for the machine and protocol used to measure muscle strength should be included.
    • Please include more details on measurement of ‘ROS’, as well as the other antioxidant/oxidative stress assays.
  • Results:
    • The statistics are difficult to compare when running different tests for different groups when you are essentially comparing all groups, especially in table form. I would consider using graphs where you can for ease of comparison. These are just suggestions in regard to concerns, as I am not an expert in this area.
    • The figure legends do not include any detail to the experiment and require much more information to understand the data in the figure/tables.
      • Additionally, there are arrows in figures 1 and 2 with no information in regards to what they mean. Likewise, the legend for figure 7 includes information about arrows, but there are no arrows in the figure.
    • Figures 4 and 5 are the exact same figure for muscle thickness, it appears that figure 5 should include muscle weight data.
    • Minor point, but many times it is stated that TCcp or oxymetholone increased body weight, calf thickness etc.; however, these data points are still below vehicle data points. I would suggest saying something to the effect of ‘TCcp attenuated the GLU-induced reductions’.

Author Response

Reviewer 1

Comments and Suggestions for Authors

The manuscript, “Tart cherry (Fruit of Prunus cerasus) concentrated powder (TCcp) ameliorates glucocorticoid-induced muscular atrophy in mice” investigates the ability of TCcp to prevent dexamethasone-induced muscle wasting and associated effects across a variety of parameters of muscle health. The evidence presented indicates that high doses of TCcp (250-500 mg/kg) is protective against dexamethasone-mediated muscle atrophy and is comparable to their positive control, oxymetholone. Overall, these data are intriguing and convincing; however, there are several issues that need to be addressed. The manuscript requires substantial revision in regards to writing and grammar for proper interpretation of these data. Additionally, several crucial details need to be added or clarified and are specified below.

General:

  1. There are several abbreviations used throughout the manuscript that are undefined.

→ To address reviewer’s comment, this has been checked throughout the manuscript. We have defined the original terms first and then used the abbreviations afterwards.

  1. The experimental design needs clarification throughout; several times it is described in a way that is confusing. For example “during the first 10 days of GLU treatment and 24 days after initial GLU treatment…” (line 210).

→ To address reviewer’s comment, we have corrected to “During 10 days of DEXA treatment and also after total 24 days of experimental periods”

  1. The abbreviation GLU is used for dexamethasone, but is also referred to as DEXA. It is unclear if the term GLU always refers to dexamethasone or to other glucocorticoids as well. It is important to be clear when referencing specific corticosteroids since glucocorticoid analogs differ greatly in their potency, half-life and specificity for the GR.

→ This is an error. We have corrected to “dexamethasone (DEXA, representative glucocorticoid; GLU)”.

  1. There is unnecessary redundancy in regards to results and other details throughout.

→ To address reviewer’s comment, we have deleted unnecessary redundancy in the revised manuscript.

  1. Several references are either out of order or incorrect making it difficult to research certain statements made by the authors, please correct.

→ To address reviewer’s comment, we have corrected the out of order or incorrect references.

  1. There are detrimental side effects of oxymetholone that should be mentioned and cited within the manuscript, including infertility and enhanced MDA levels in mice. Future studies should investigate whether there are similar compounds within TCcp and examine whether similar side effects exist with this botanical extract.

→ To address reviewer’s comment, we have added following sentences in the Discussion and Reference sections.

De Souza and Hallak (2011) showed that long-term, non-therapeutic doses of oxymetholone can increase atrophy and reduce germinal cells in the testicles and also change the serum levels of LH, FSH hormones by increasing free radicals and changing serum testosterone levels (De Souza & Hallak, 2011). Further reported detrimental side effects of oxymetholone include infertility and enhanced MDA levels in mice (Akbari Bazm et al., 2020; Akbari Bazm et al., 2019).

De Souza, G.L.; Hallak, J. Anabolic steroids and male infertility: A comprehensive review. BJU Int 2011, 108, pp. 1860–1865.

Akbari Bazm, M.; Goodarzi, N.; Shahrokhi, S.R.; Khazaei, M. The effects of hydroalcoholic extract of Vaccinium arctostaphylos L. on sperm parameters, oxidative injury and apoptotic changes in oxymetholone‐induced testicular toxicity in mouse. Andrologia 2020, 52, e13522.

Akbari Bazm, M.; Khazaei, M.; Khazaei, F.; Naseri, L. Nasturtium Officinale L. hydroalcoholic extract improved oxymetholone‐induced oxidative injury in mouse testis and sperm parameters. Andrologia 2019, 51, e13294.

After searching for similar compounds of oxymetholone in Prunus cerasus, using various online databases (Web of Science, Science Direct, MEDLINE, and PubMed), no results were found, that we plan to study in our upcoming research. As the reviewer suggested, future studies could be required to elucidate any similar compounds in TCcp and to examine whether similar side effects exist within this botanical extract.

  1. It would be nice to see some western data to support the mRNA findings as mRNA expression does not always coincide with translation of protein. This would be especially useful when evaluation pathways such as Akt and other proteins whose activity and function depends on posttranslational modifications and less on amount of expression.

→ I fully agree with your opinion that mRNA expression does not always coincide with protein translation. If there is an opportunity in the future, we would like to confirm the enhancement of Akt1 mRNA expression and protein translation by TCcp. Due to limited resources for the current study/project; this could be performed in a future study.

  1. It appears that only male mice were used for these experiments, but that is not clearly stated. Please clarify and consider including female mice in future studies.

→ To address reviewer’s comment, we have added "male" in 2.2. Animal husbandry and experimental design and male ICR mice. Due to limited resources, it is not possible to do the same study on the female mice; however, this could be performed in a future study.

  1. Future studies should consider evaluation more than one muscle group and including NMR/MRI to evaluate changes in total lean mass, although when focusing on one the gastrocnemius is a good choice for several reasons.

→ We have tried thicknesses of both calf and gastrocnemius muscle measurements in this study. Thank you for your suggestion, we plan to evaluation more than one muscle group and including NMR/MRI to evaluate changes in total lean mass in a future study.

Methods:

  1. SPF ICR mice- the abbreviations are not noted. Please explain why you chose this strain for these studies.

→ To address reviewer’s comment, we have revised to "Specific pathogen free (SPF) Institute of Cancer Research (ICR) mice".

  1. The catalog number for TCcp needs to be included and the information for the vehicle needs clarification. For example, ‘Pink TCcp powder was dissolved at a concentration of 50mg/mL in distilled water and used as vehicle’ (line 89-90) is quite confusing.

→ To avoid misunderstanding, we have revised the sentence to " Pink TCcp powder (Anderson Global Group; Irvine, CA, USA) was dissolved at a concentration of 50 mg/mL in distilled water and used as a vehicle.

  1. It seems that all groups with the exclusion of vehicle are given GLU, but this was not clearly stated and could be confusing to the reader.

→ The following information has already been mentioned in the manuscript "Different doses of TCcp such as 500, 250, and 125 mg/kg were orally supplied (in a volume of 10 mL/kg) once daily using a zonde attached 1 mL syringe for 24 days starting from two weeks before DEXA treatment."

  1. It is unclear whether the entire triceps surae was measured or the soleus was removed and just the gastrocnemius was assessed for weight and thickness. I would suggest avoiding the use of “calf”.

→ The thicknesses of calf and gastrocnemius muscle was measured during this study as stated in the manuscript.

  1. The details for the machine and protocol used to measure muscle strength should be included.

→ To address reviewer’s comment, we have added the following sentences in the revised manuscript.

Calf muscle strength measurements

From 1 h after last (24th administration of vehicle) treatment, oxymetholone or all three different dosages of TCcp 500, 250 and 125 mg/kg, the calf muscle strengths of individual mice were measured as tensile strengths (Newton, N) using a computerized testing machine (Japan Instrumentation System Co., Tokyo, Japan) by following the previously established methods (Kim et al., 2015a,b). Briefly, animals were restrained into machines using two separated 1-0 silk suture ties on left ankle and chest, and the peak tensile loads were recorded as calf muscle strengths, during knee angles reach 0° (10 ~ 20 mm distance).

  1. Kim, J.W.; Ku, S.K.; Han, M.H.; Kim, K.Y.; Kim, S.G.; Kim, G.Y.; Hwang, H.J.; Kim, B.W.; Kim, C.M.; Choi, Y.H. The administration of Fructus Schisandrae attenuates dexamethasone-induced muscle atrophy in mice. Int J Mol Med 2015a, 36, pp. 29-42.

  1. Please include more details on measurement of ‘ROS’, as well as the other antioxidant/oxidative stress assays.

→ The concentrations of gastrocnemius muscles lipid peroxidation were determined by estimating MDA via thiobarbituric acid test at 525 nm absorbance using UV/Vis spectrophotometer (Mecasys, Daejeon, Rep. of Korea). The results were presented as nM of MDA/g tissue [Jamall and Smith, 1985] [46]. ROS level analysis was performed using 2’,7’-dichlorofluorescein diacetate (DCFDA) fluorescent dye as a probe and fluorescence density was measured at 490/520 nm according to the manufacturer’s guidelines of Cel-lular Reactive Oxygen Species Detection Assay Kit (Abcam, Cambridge, MN, USA). The measured values of optical density [a relative fluorescence unit (RFU)] were corrected by the protein concentrations of samples and were expressed as RFU/μg protein [He et al., 2012] [47]. GSH content were measured at 412 nm absorbance of 2-nitrobenzoic acid (Sigma-Aldrich, St. Louise, MO, USA) and the results were expressed as mg/g tissue [Sedlak and Lindsay, 1968] [48]. Decomposition of H2O2 in the presence of catalase was followed at 240 nm [Aebi et al., 1974] [49]. SOD estimation was based on the generation of superoxide radicals produced by xanthine and xanthine oxidase, which react with nitro-tetrazolium blue to form formazan dye. SOD activity was measured at 560 nm and was expressed as U/mg protein.”

Results:

  1. The statistics are difficult to compare when running different tests for different groups when you are essentially comparing all groups, especially in table form. I would consider using graphs where you can for ease of comparison. These are just suggestions in regard to concerns, as I am not an expert in this area.

→ I am grateful for your comment. This manuscript contains a lot of data. When using Figures, the volume of the manuscript becomes too large (almost 30 pages), thus we used tables to represent the experimental data. I hope for your kind understanding.

  1. The figure legends do not include any detail to the experiment and require much more information to understand the data in the figure/tables.

→ We totally agree and respect with your opinion and suggestion. However, if the legend of each figure and table contains detailed information about the experiment, the overall manuscript volume becomes too large (almost 30 pages). We would like our readers to find materials and methods section to check the details of the experiment. Please accept our apology for not reflecting your suggestion in the revised manuscript, and we ask for your kind understanding.

  1. Additionally, there are arrows in figures 1 and 2 with no information in regards to what they mean. Likewise, the legend for figure 7 includes information about arrows, but there are no arrows in the figure.

→ To address reviewer’s comment, we have added the following sentences:

Figure 1.

Dot arrows show significant decreases in body weights in GLU control mice after 5 days of initial DEXA treatment to sacrifice. However, these decreases of body weights were significantly inhibited by treatment of oxymetholone, TCcp 500 and 250 mg/kg, respectively (arrowhead).

Day -1 and Day-24 means 1 day before start of test material administration and at sacrifice, respectively.

Day 0 means at start of test material administration, at 2 weeks before initial DEXA treatment.

All animals were overnight fasted before first test material administration and sacrifice (arrows).

Figure 2.

Dot arrows show significant decreases in calf thickness in GLU control mice after 5 days of initial DEXA treatment to sacrifice as compared with the intact control. However, these decreases in calf thicknesses were significantly and dose-dependently inhibited by oxymetholone, TCcp 500 and 250 mg/kg treatments, respectively (arrowhead).

Day -1 and Day-24 means 1 day before start of test material administration and at sacrifice, respectively.

Day 0 means at start of test material administration, at 2 weeks before initial DEXA treatment.

All animals were overnight fasted before first test material administration and sacrifice (arrows).

  1. Figures 4 and 5 are the exact same figure for muscle thickness; it appears that figure 5 should include muscle weight data.

→ This was a mistake. We have changed Figure 5 and we apologize for any inconvenience caused.

  1. Minor point, but many times it is stated that TCcp or oxymetholone increased body weight, calf thickness etc.; however, these data points are still below vehicle data points. I would suggest saying something to the effect of ‘TCcp attenuated the GLU-induced reductions’.

→ To address reviewer’s comment, we have changed this information to “TCcp attenuated the GLU-induced reductions” throughout the revised manuscript.

Thank you for your valuable comments on our manuscript. We have addressed and corrected the defects you point out, and we ask that the revised manuscript be re-examined and reconsidered for publication.

Reviewer 2 Report

Thank you for submitting your paper for review. The present study assessed the possible favorable effects of tart cherry concentrated powders (TCcp) on skeletal muscular preservation using the DEXA induced muscular atrophy model in mice. The authors investigated whether administration of TCcp can prevent or improve catabolic DEXA induced muscular atrophy, and the possible mechanisms, using different concentration of TCcp.  

I do have some concerns that I believe must be addressed, which I have detailed below for your consideration. I hope these comments help to improve the manuscript.

Major:

How the mice were sacrificed? How and from blood was taken from mice? There is also no information on how the homogenates or supernatants from gastrocnemius muscle were prepared, and what method was used to determine the protein content in muscle. There are no details related to "Antioxidant analysis" such as: what kind of methods, apparatus, chemicals, No and buffers, etc. were used. Please add these details to M&M section.

It is well known that very high activity of both CK and LDH enzymes in the serum, indicates “muscle damage”.  However, the results presented in Table 3 are in opposite direction. Moreover, the authors tried to explain these controversial data as followed:

Line: 593 “Plasma levels of LDH and CK are used as markers of muscle injury [26], and are markedly elevated in disused muscular atrophic animals [54]. In animals with catabolic muscular atrophy induced by GLU treatment, marked increases in levels of serum CK were observed; however, levels of serum LDH normally decrease because of reduced contraction of skeletal muscle fiber and physiological activities [26]. Significant decreases in serum LDH levels, indicating reduced muscular activity, and increases in serum CK levels, suggesting muscular damage, were also observed in GLU control”  in the discussion section.

I do not accept this kind of explanation, what's more, with wrong chosen citations. Please change the explanation with corrected citations.  

The bibliography is inappropriately cited, for example:

Line 132: “One hour after the 24th administration of TCcp, oxymetholone, or vehicle, the muscle strength of calf in each mouse was calculated as tensile strength by an automated testing machine (Japan Instrumentation System Co., Tokyo, Japan) as demonstrated by Kim et al. [26].

Line 593: “Plasma levels of LDH and CK are used as markers of muscle injury [26], and are markedly elevated in disused muscular atrophic animals [54]. In animals with catabolic muscular atrophy induced by GLU treatment, marked increases in levels of serum CK were observed; however, levels of serum LDH normally decrease because of reduced contraction of skeletal muscle fiber and physiological activities [26]”.

Line 646: “Furthermore, significant increase in atrogin-1 and MuRF1 mRNA expression has been noticed in GLU-646 induced catabolic atrophy [16, 26, 28]”.

Line 658: “Marked down regulated mRNA expressions of Akt1 and PI3K were observed in GLU control, indicating changes in catabolic muscular atrophy, which is similar with the findings of a previous study [26]”.

Line 688: “During catabolic muscular atrophy, GLU treatments have reported decreased mRNA expression of SIRT1 and myostatin and muscular mass [16, 25, 26]”.

How many times can be cited the same reference in different aspects?

“26. Dorfman, R.I.; Kincl, F.A. Relative potency of steroids in an anabolic–androgenic assay using the castrated rat. Endocrinology 1963, 72, pp. 259–266”.

“28.Young, G.P.; Bhathal, P.S.; Sullivan, J.R.; Wall, A.J.; Fone, D.J.; Hurley, T.H. Fatal hepatic coma complicating oxymetholone therapy in multiple myeloma. Aust N Z J Med 1977, 7, pp. 47–51”.

“33. Kim, D.O.; Heo, H.J.; Kim, Y.J.; Yang, H.S.; Lee, C.Y. Sweet and sour cherry phenolics and their protective effects on neuronal cells. J Agric Food Chem 2005, 53, pp. 9921–9927”.

Please check the whole manuscript and use appropriate citations, and arrange the references according to citations and related to the true context.

Minor:

In the manuscript, there are typos, grammar, and other errors.

For example:

Abstract:

Line: 21 first was used GLU and explanation there is in Line: 22

Line: 25 did the authors measure CK and LDH contents? No

Line: 392 sirtulin … should be sirtuin

Line: 437 GLC control mice … what means GLC?

Line: 543 Atrophy ….

Line: 586  However……

No at all explanation for abbreviation used in the manuscript.

Please, check the whole body text for these kinds of mistakes and correct them, please.

Author Response

Reviewer 2

Comments and Suggestions for Authors

Thank you for submitting your paper for review. The present study assessed the possible favorable effects of tart cherry concentrated powders (TCcp) on skeletal muscular preservation using the DEXA induced muscular atrophy model in mice. The authors investigated whether administration of TCcp can prevent or improve catabolic DEXA induced muscular atrophy, and the possible mechanisms, using different concentration of TCcp. I do have some concerns that I believe must be addressed, which I have detailed below for your consideration. I hope these comments help to improve the manuscript.

Major:

  1. How the mice were sacrificed? How and from blood was taken from mice?

→ To address reviewer’s comment, we have added “Animals were sacrificed by cervical dislocation.” and “Approx. 200 μL of whole blood sample was drawn from the posterior vena cava using a syringe with a 26-gauge needle under 2-3% isofurane (Hana Pharm., Hwasung, Rep. of Korea) inhalation anesthesia.”

  1. There is also no information on how the homogenates or supernatants from gastrocnemius muscle were prepared, and what method was used to determine the protein content in muscle.

→ In response to your comment, we have added the following text:

Methods

 “Separated gastrocnemius muscles were weighed, homogenized in ice-cold 0.01M Tris-HCl (pH 7.4), and centrifuged at 12,000× g for 15 min as described by Del Rio et al. [2005] [44]. Total protein content were measured by following the previously reported method [Lowry et al., 1951] [45] and bovine serum albumin (Invitrogen, Carlsbad, CA, USA) as an internal standard.”

References

“44. Del Rio, D., Stewart, A.J.; Pellegrini, N. A review of recent studies on malondialdehyde as toxic molecule and biological marker of oxidative stress. Nutr Metab Cardiovasc Dis 2005,15, pp. 316-328.

  1. Lowry, O.H.; Rosenbrough, N.J.; Farr, A.L.; Randall, R.J. Protein measurement with the Folin phenol reagent. J Biol Chem 1951, 193, pp. 265-275.”

  1. There are no details related to "Antioxidant analysis" such as: what kind of methods, apparatus, chemicals, No and buffers, etc. were used. Please add these details to M&M section.

→ To address the reviewer’s comments, we have added the following text in the revised manuscript.

“The concentrations of gastrocnemius muscles lipid peroxidation were determined by estimating MDA via thiobarbituric acid test at 525 nm absorbance using UV/Vis spectrophotometer (Mecasys, Daejeon, Rep. of Korea). The results were presented as nM of MDA/g tissue [Jamall and Smith, 1985] [46]. ROS level analysis was performed using 2’,7’-dichlorofluorescein diacetate (DCFDA) fluorescent dye as a probe and fluorescence density was measured at 490/520 nm according to the manufacturer’s guidelines of Cel-lular Reactive Oxygen Species Detection Assay Kit (Abcam, Cambridge, MN, USA). The measured values of optical density [a relative fluorescence unit (RFU)] were corrected by the protein concentrations of samples and were expressed as RFU/μg protein [He et al., 2012] [47]. GSH content were measured at 412 nm absorbance of 2-nitrobenzoic acid (Sigma-Aldrich, St. Louise, MO, USA) and the results were expressed as mg/g tissue [Sedlak and Lindsay, 1968] [48]. Decomposition of H2O2 in the presence of catalase was followed at 240 nm [Aebi et al., 1974] [49]. SOD estimation was based on the generation of superoxide radicals produced by xanthine and xanthine oxidase, which react with nitro-tetrazolium blue to form formazan dye. SOD activity was measured at 560 nm and was expressed as U/mg protein.”

Jamall, I.S.; Smith, J.C. Effects of cadmium on glutathione peroxidase, superoxidase dismutase and lipid peroxidation in the rat heart: a possible mechanism of cadmium cardiotoxicity. Toxicol Appl Pharmacol 1985, 80, pp. 33-42.

He, H.J.; Wang, G.Y.; Gao, Y.; Ling, W.H.; Yu, Z.W.; Jin, T.R. Curcumin attenuates Nrf2 signaling defect, oxidative stress in muscle and glucose intolerance in high fat diet-fed mice. World J Diabetes 2012, 3, pp. 94-104.

Sedlak, J.; Lindsay, R.H. Estimation of total, protein-bound, and nonprotein sulfhydryl groups in tissue with Ellman's reagent. Anal Biochem 1968, 25, pp. 192-205.

Aebi H. Catalase. In: Methods in Enzymatic Analysis; Bergmeyer, H.U. Eds.; Academic Press: New York, USA, 1974; pp. 673-686.

  1. It is well known that very high activity of both CK and LDH enzymes in the serum, indicates “muscle damage”. However, the results presented in Table 3 are in opposite direction. Moreover, the authors tried to explain these controversial data as followed:

Line: 593 “Plasma levels of LDH and CK are used as markers of muscle injury [26], and are markedly elevated in disused muscular atrophic animals [54]. In animals with catabolic muscular atrophy induced by GLU treatment, marked increases in levels of serum CK were observed; however, levels of serum LDH normally decrease because of reduced contraction of skeletal muscle fiber and physiological activities [26]. Significant decreases in serum LDH levels, indicating reduced muscular activity, and increases in serum CK levels, suggesting muscular damage, were also observed in GLU control” in the discussion section.

→ In general, it is correct that very high activity of LDH enzymes in the serum indicates “muscle damage”, as you pointed out. However, significant elevations of serum CK levels indicating muscle damages and decreases of serum LDH levels suggesting reduction of muscle activities were also demonstrated in GLU control mice in the present experiment. It is consistent with the previous reports by Orzechowski et al. (2002), Pellegrino et al. (2004), and Kim et al. (2015a) stating that DEXA treatment in catabolic muscular atrophic animals shows marked elevations of serum CK levels, but serum LDH levels were generally decreased due to reduction in the physiological activities such as the contractions of skeletal muscle fiber [Orzechowski et al., 2002; Pellegrino et al., 2004; Kim et al., 2015].

→ In animals with catabolic muscular atrophy induced by DEXA treatment, marked increases in serum CK levels were observed; however, serum LDH levels decreased due to reduced physiological activities such as contraction of skeletal muscle fiber [59, 60].”

Orzechowski, A.; Ostaszewski, P.; Wilczak, J.; Jank, M.; Bałasińska, B.; Wareski, P.; Fuller, J. Jr. Rats with a glucocorticoid-induced catabolic state show symptoms of oxidative stress and spleen atrophy: the effects of age and recovery. J Vet Med A Physiol Pathol Clin Med 2002, 49, pp. 256-263.

Pellegrino, M.A.; D'Antona, G.; Bortolotto, S.; Boschi, F.; Pastoris, O.; Bottinelli, R.; Polla, B.; Reggiani, C. Clenbuterol antagonizes glucocorticoid-induced atrophy and fibre type transformation in mice. Exp Physiol 2004, 89, pp. 89-100.

I do not accept this kind of explanation, what's more, with wrong chosen citations. Please change the explanation with corrected citations. 

The bibliography is inappropriately cited, for example:

  1. Line 132: “One hour after the 24th administration of TCcp, oxymetholone, or vehicle, the muscle strength of calf in each mouse was calculated as tensile strength by an automated testing machine (Japan Instrumentation System Co., Tokyo, Japan) as demonstrated by Kim et al. [26].

→ To address reviewer’s comment, we have extensively revised the entire manuscript and the references have been corrected accordingly.

[43] Kim, J.W.; Ku, S.K.; Han, M.H.; Kim, K.Y.; Kim, S.G.; Kim, G.Y.; Hwang, H.J.; Kim, B.W.; Kim, C.M.; Choi, Y.H. The administration of Fructus Schisandrae attenuates dexamethasone-induced muscle atrophy in mice. Int J Mol Med 2015a, 36, pp. 29-42.

  1. Line 593: “Plasma levels of LDH and CK are used as markers of muscle injury [26], and are markedly elevated in disused muscular atrophic animals [54]. In animals with catabolic muscular atrophy induced by GLU treatment, marked increases in levels of serum CK were observed; however, levels of serum LDH normally decrease because of reduced contraction of skeletal muscle fiber and physiological activities [26]”.

→ To address the reviewer’s comments, we have tried

  1. Line 646: “Furthermore, significant increase in atrogin-1 and MuRF1 mRNA expression has been noticed in GLU-646 induced catabolic atrophy [16, 26, 28]”.

→ To address the reviewer’s comments, we have tried

  1. Line 658: “Marked down regulated mRNA expressions of Akt1 and PI3K were observed in GLU control, indicating changes in catabolic muscular atrophy, which is similar with the findings of a previous study [26]”.

→ To address reviewer’s comment, we have corrected the reference [43].

  1. Line 688: “During catabolic muscular atrophy, GLU treatments have reported decreased mRNA expression of SIRT1 and myostatin and muscular mass [16, 25, 26]”.

→ To address the reviewer’s comments, we have tried

  1. How many times can be cited the same reference in different aspects?

“26. Dorfman, R.I.; Kincl, F.A. Relative potency of steroids in an anabolic–androgenic assay using the castrated rat. Endocrinology 1963, 72, pp. 259–266”.

“28. Young, G.P.; Bhathal, P.S.; Sullivan, J.R.; Wall, A.J.; Fone, D.J.; Hurley, T.H. Fatal hepatic coma complicating oxymetholone therapy in multiple myeloma. Aust N Z J Med 1977, 7, pp. 47–51”.

“33. Kim, D.O.; Heo, H.J.; Kim, Y.J.; Yang, H.S.; Lee, C.Y. Sweet and sour cherry phenolics and their protective effects on neuronal cells. J Agric Food Chem 2005, 53, pp. 9921–9927”.

Please check the whole manuscript and use appropriate citations, and arrange the references according to citations and related to the true context.

→ To address reviewer’s comment, we have extensively revised the entire manuscript and the references have been corrected accordingly.

Minor:

  1. In the manuscript, there are typos, grammar, and other errors.

For example:

Abstract:

Line: 21 first was used GLU and explanation there is in Line: 22

Line: 25 did the authors measure CK and LDH contents?

Line: 392 sirtulin … should be sirtuin.

Line: 437 GLC control mice … what means GLC? → GLU is correct.

Line: 543 Atrophy ….

Line: 586 However……

→ To address reviewer’s comments, The pointed mistakes have been corrected. Furthermore, the manuscript was rechecked by a native speaker for any possible grammar or spelling mistake.

  1. No at all explanation for abbreviation used in the manuscript.

→ To address reviewer’s comment, this has been checked throughout the manuscript. We have defined the original terms first and then used the abbreviations afterwards.

  1. Please, check the whole body text for these kinds of mistakes and correct them, please.

→ Thank you for your kind suggestions, we have tried our best to cover all your comments and we have corrected the manuscript accordingly.

Thank you for your valuable comments on our manuscript. We have addressed and corrected the defects you point out, and we ask that the revised manuscript be re-examined and reconsidered for publication.

Round 2

Reviewer 1 Report

Significant improvements have been made to the manuscript "Tart cherry (Fruit of Prunus cerasus) concentrated powder (TCcp) ameliorates glucocorticoid-induced muscular atrophy in mice" and the majority of my concerns were addressed. These data are compelling and have merit. I appreciate the implemented revisions; however, some important issues remain and need to be corrected. For example, there are still several grammatical errors throughout the text. Furthermore, upon randomly checking a few references, some citations still do no match with the given statement, I would recommend using an automated reference manager to help with this if this is available -- Mendeley is a great free option in the US. Lastly, I understand the authors' desire to keep the page numbers to a minimum; however, I do not think cutting out basic information regarding the experimental details for the figure is the answer. As a reader of scientific literature, I find it frustrating to have to refer to the methods section to understand what is being displayed every time I see a figure. Providing minimal detail here would be quite beneficial in my opinion. Along these lines, the authors have added data interpretation to the legends figures 1 and 2--this information does not belong here. The legend provided in Figure 2 is nice (aside from the inclusion of data interpretation). I would suggest editing the other figures to resemble this and removing the identification of all acronyms that have already been described previously in every figure legend to save space.

Author Response

Responding letter to the reviewer’s comments

Reviewer 1

Significant improvements have been made to the manuscript "Tart cherry (Fruit of Prunus cerasus) concentrated powder (TCcp) ameliorates glucocorticoid-induced muscular atrophy in mice" and the majority of my concerns were addressed. These data are compelling and have merit. I appreciate the implemented revisions; however, some important issues remain and need to be corrected.

  1. For example, there are still several grammatical errors throughout the text.

→ To address reviewer’s comment, this has been checked throughout the manuscript.

  1. Furthermore, upon randomly checking a few references, some citations still do no match with the given statement, I would recommend using an automated reference manager to help with this if this is available -- Mendeley is a great free option in the US.

→ Thank you for your valuable comment. To address reviewer’s comment, we have corrected the incorrect references.

  1. Lastly, I understand the authors' desire to keep the page numbers to a minimum; however, I do not think cutting out basic information regarding the experimental details for the figure is the answer. As a reader of scientific literature, I find it frustrating to have to refer to the methods section to understand what is being displayed every time I see a figure. Providing minimal detail here would be quite beneficial in my opinion. Along these lines, the authors have added data interpretation to the legends figures 1 and 2--this information does not belong here. The legend provided in Figure 2 is nice (aside from the inclusion of data interpretation). I would suggest editing the other figures to resemble this and removing the identification of all acronyms that have already been described previously in every figure legend to save space.

→ I fully agree with your opinion. To address reviewer’s comment, we have added minimal details in the Figure and Table legends.

Thank you for your valuable comments on our manuscript. We have addressed and corrected the defects you point out, and we ask that the revised manuscript be re-examined and reconsidered for publication.

Reviewer 2 Report

The authors almost addressed my comments and suggestions. However, there still exist mistakes in the manuscript.

M&M section:

Line 210-221; the citations are not corrected. The work according to Dorfman and Kincl was on the accuracy of rat androgenic-anabolic methods using either subcutaneous or oral routes of administration (ref. 26).

Could you check again the references list and citations in the whole body text, please. I am afraid that there are other -uncorrected citations.

In the manuscript, there are typo errors.

Author Response

Reviewer 2.

Comments and Suggestions for Authors

The authors almost addressed my comments and suggestions. However, there still exist mistakes in the manuscript.

M&M section:

  1. Line 210-221; the citations are not corrected. The work according to Dorfman and Kincl was on the accuracy of rat androgenic-anabolic methods using either subcutaneous or oral routes of administration (ref. 26).

→ This is an error. We have corrected to [43] from [26].

  1. Could you check again the references list and citations in the whole body text, please. I am afraid that there are other -uncorrected citations.

→ To address reviewer’s comment, we have corrected the incorrect references.

  1. In the manuscript, there are typo errors.

→ To address reviewer’s comment, this has been checked throughout the manuscript.

Thank you for your valuable comments on our manuscript. We have addressed and corrected the defects you point out, and we ask that the revised manuscript be re-examined and reconsidered for publication.